# Marine and terrestrial contributions to atmospheric deposition fluxes of methylated arsenic species

Esther S. Breuninger [1,2] ✉, Julie Tolu [1,2] ✉, Franziska Aemisegger[3,4,5], Iris Thurnherr[3], Sylvain Bouchet [1,2], Adrien Mestrot[4,5], Rachele Ossola[2,6], Kristopher McNeill[2], Dariya Tukhmetova[7], Jochen Vogl [7], Björn Meermann [7], Jeroen E. Sonke [8] & Lenny H. E. Winkel [1,2] ✉

Arsenic, a toxic element from both anthropogenic and natural sources, reaches surface environments through atmospheric cycling and dry and wet deposition. Biomethylation volatilizes arsenic into the atmosphere and deposition cycles it back to the surface, affecting soil-plant systems. Chemical speciation of deposited arsenic is important for understanding further processing in soils and bioavailability. However, the range of atmospheric transport and source signature of arsenic species remain understudied. Here we report significant levels of methylated arsenic in precipitation, cloud water and aerosols collected under free tropospheric conditions at Pic du Midi Observatory (France) indicating long-range transport, which is crucial for atmospheric budgets. Through chemical analyses and moisture source diagnostics, we identify terrestrial and marine sources for distinct arsenic species. Estimated atmospheric deposition fluxes of methylated arsenic are similar to reported methylation rates in soils, highlighting atmospheric deposition as a significant, overlooked source of potentially bioavailable methylated arsenic species impacting plant uptake in soils.

Arsenic (As) has been classified as a Group 1 carcinogen by the International Agency for Research and Cancer (IARC)[1]. Toxicity of As varies between inorganic and organic species, and is generally higher for trivalent (oxidation state III) than pentavalent (oxidation state V) species[2]. With respect to environmental As exposure, most studies focused on As contamination of groundwater and (paddy) soils, whereas other environmental compartments such as the atmosphere have received less attention. However, the atmosphere is a significant and dynamic reservoir of As, in which an estimated 31 Gg of As is annually cycled[3] and can represent an important source of As to

surface ecosystems and food chains[4,5]. Given its estimated lifetime of 4 to 5 days, As can also be transported over longer distances by large-scale weather systems before being removed from the atmosphere by mainly wet deposition (estimated at 82% of total deposition)[3]. Therefore, not only locally emitted atmospheric As but also As emitted from distant sources is important for the pool of atmospheric As deposited to surface environments, however, deposition from long-range sources is largely unstudied.

Global inventories have estimated that the majority of atmospheric As is of anthropogenic origin, with estimated contributions of

[1]Eawag, Swiss Federal Institute of Aquatic Science and Technology, Dübendorf, Switzerland. [2]Institute of Biogeochemistry and Pollutant Dynamics, ETH Zurich, Zurich, Switzerland. [3]Institute for Atmospheric and Climate Science, ETH Zurich, Zurich, Switzerland. [4]Institute of Geography, University of Bern, Bern, Switzerland. [5]Oeschger Centre for Climate Change Research, University of Bern, Bern, Switzerland. [6]Department of Chemistry, Colorado State University, Fort Collins, Colorado, USA. [7]Federal Institute for Materials Research and Testing, Division 1.1—Inorganic Trace Analysis, Berlin, Germany. [8]Géosciences Environnement Toulouse, CNRS/IRD/Université de Toulouse, Toulouse, France. ✉e-mail: esther.breuninger@usys.ethz.ch; julie.tolu@eawag.ch; lenny.winkel@eawag.ch

93% of total emissions[3]. Anthropogenic emissions include mining, smelting of non-ferrous metals, landfills, and burning of fossil fuels[6], which are expected to predominately emit inorganic As (e.g. as arsenic trioxide: $As^{III}_2O_3$ or arsenate: $As^VO_4^{3-}$)[7,8]. Due to air pollution control, anthropogenic emissions have decreased in the last decades[9], particularly in North America and Europe, which will continuously increase the relative importance of environmental emissions of legacy As via biogenic processes, similar to mercury[10]. Such biogenic processes include the (bio) transformation and volatilisation of inorganic As by the addition of a methyl group ($CH_3$) forming the species monomethylarsine ($CH_3As^{III}H_2$), dimethylarsine (($CH_3$)$_2As^{III}H$) and trimethylarsine (($CH_3$)$_3As^{III}$) by various fungi, bacteria, and algae[11], with trimethylarsine being the predominant species emitted from paddy soils[12] and seawater incubation experiments[13,14]. Other natural sources of (methylated) As include wind erosion of soil, volcanism, and geothermal emissions[15]. Volcanic environments can emit arsine ($AsH_3$) and small amounts of methylated arsines[16].

Volatile As has an estimated half-life of ~8 h (day-time conditions) for methylarsines and even longer for inorganic arsines and in the dark[17]. Gaseous species are expected to transform into non-volatile, oxidized pentavalent (oxidation state V) compounds, including arsenate ($As^V$), as well as oxidized methylated As species such as monomethylarsonic acid ($CH_3As^VO_3H_2$, abbreviation used thereafter: "MMAs$^V$"), dimethylarsinic acid (($CH_3$)$_2As^VO_2H$, "DMAs$^V$") and trimethylarsine oxide (($CH_3$)$_3As^VO$, "TMAs$^V$O"), all of which are expected to undergo gas-particle transformation[17,18]. While few studies have investigated the occurrence of methylated As species in atmospheric deposition[18–24], the degree to which long-range transport of variable sources can contribute to their deposition is still unclear. Previous studies aimed to investigate local to regional sources by sampling in (sub-)urban/rural[18,20–24] or forest areas[19]. To specifically study long-range transport, locations that are frequently exposed to free tropospheric air with limited influence of local processes (such as at high altitude) are most suitable. Furthermore, the source apportionment of As species in atmospheric deposition and their quantitative impacts on the terrestrial environment are still poorly constrained. Beyond informing about atmospheric As sources, knowing the chemical forms (speciation) of As in wet and dry deposition is essential because it determines the fate of deposited As in surface environments, e.g. its mobility, bioavailability for plant uptake and/or toxicity. For example, DMAs$^V$ (possibly carcinogenic)[1] has a higher accumulation rate in rice plants due to more efficient transfer from roots to the grain than inorganic As species[25], increasing the potential dietary exposure risk.

Here, we investigate the concentrations and speciation of methylated As$^V$, as a tracer of biogenic emissions, in atmospheric samples collected at the high-altitude monitoring station Pic du Midi Observatory (French Pyrenees; 2877 m a.s.l.). This station is mostly exposed to the free troposphere, thus allowing the investigation of long-range elemental transport from both continental (Europe, Africa) and marine (Atlantic Ocean and Mediterranean Sea) regions[26], in addition to transport from local sources. We collected a series of precipitation and cloud water samples at high temporal resolution (sub-event-based) during a 2-month campaign in 2019. In combination with air parcel backward trajectories and Lagrangian moisture source analyses, this sampling approach allowed us to link As speciation to moisture sources that contributed to wet deposition at Pic du Midi, and to identify dominant atmospheric transport patterns of As species. To further constrain the contributing source signatures of atmospheric As species, we collected weekly aerosol samples between 2015 and 2020 and characterized various chemical proxies, including total concentrations of (trace) elements, sulfur (S) isotopes and speciation, and organic aerosol composition. Using this suite of samples and analytical measurements, we could accurately quantify the full range of methylated species in an environment receiving atmospheric deposition from a broad range of sources, i.e. from local to distant, terrestrial to marine and anthropogenic to biogenic. Our data demonstrate that biogenic-derived As is quantitatively important even under such heterogeneous source conditions, implying that methylated As species should be included in estimates of atmospheric budgets. Furthermore, through detailed atmospheric analyses and linking As speciation to chemical proxy data, we could identify differences in sources of methylated As species. Finally, by comparing our atmospheric data to previously measured As methylation rates in soil pore waters, we show that atmospheric deposition fluxes of methylated As species are likely a significant source of these species to surface environments.

## Results & discussion
### Concentrations of total As and As species in wet deposition
Total As concentrations in precipitation (sub)events ("P" samples) range from 0.010 to 0.074 $\mu g \cdot L^{-1}$, resulting in total As deposition of 0.007–0.270 $\mu g \cdot m^{-2}$ ($n = 26$, median: 0.038± $\mu g \cdot m^{-2}$; Fig. 1a, Supplementary Fig. 1c and Supplementary Table 1). In cloud water ("C" samples), total As concentrations range from 0.026–0.441 $\mu g \cdot L^{-1}$, corresponding to 0.001–0.009 $ng \cdot m^{-3}$ ($n = 56$ sampled during 2019 campaign, median: 0.002 $ng \cdot m^{-3}$; Fig. 1b and Supplementary Fig. 1b). Concentrations of As quantified in our study are much lower than previously reported ranges for precipitation collected in an urban area (Belfast; 0.045–4.5 $\mu g \cdot L^{-1}$)[21] and a forest (Fichtelgebirge, Germany; 0.137–1210 $\mu g \cdot L^{-1}$)[19] as well as for cloud water taken at mountain sites in Germany, France, Pakistan, China and the USA (0.1–115 $\mu g \cdot L^{-1}$)[27–31]. The generally lower concentrations in wet deposition samples collected at Pic du Midi are expected because of its remote location and its elevation. This site is mostly exposed to free tropospheric air with limited influences from the planetary boundary layer[32], although rainout along the pathway to the high-altitude site might reduce contributions from long-range transport.

To determine As speciation at these ultra-trace levels, we preconcentrated precipitation and cloud water samples by lyophilisation prior to analysis by HPLC-ICP-MS/MS (detailed description of method optimization in Supplementary Discussions 1, 2, Supplementary Figs. 2, 3 and Table 2). Our optimized procedure reaches a detection limit of 1-2 $ng \cdot L^{-1}$ depending on the As species, which is 4 to 25 times lower than methods previously applied to precipitation[19–21] and aerosol[22–24] samples. This improvement in detection limits allow the quantification of TMAs$^V$O and DMAs$^V$ in all collected precipitation samples in contrast to most previous studies (DMAs$^V$ being below or equal to detection limits in >99% [19–21] and TMAs$^V$O in >67% of the samples analysed in previous studies[19,21]). Due to the instability of reduced inorganic As (As$^{III}$) in precipitation and applied aerosol extraction methods (described in next section), As$^{III}$ and As$^V$ are determined as total inorganic arsenic (iAs). Concentrations in precipitation range from 1 to 3 $ng \cdot L^{-1}$ for DMAs$^V$ (median of all sub-events: 1 $ng \cdot L^{-1}$), from 2 to 24 $ng \cdot L^{-1}$ for TMAs$^V$O (median: 4 $ng \cdot L^{-1}$), and from 6 to 52 $ng \cdot L^{-1}$ for iAs (median: 18 $ng \cdot L^{-1}$). These concentrations correspond to deposition ranges of 1–13 $ng \cdot m^{-2}$, 2–52 $ng \cdot m^{-2}$ and 4–194 $ng \cdot m^{-2}$ for DMAs$^V$ and TMAs$^V$O and iAs (Fig. 1a; Supplementary Table 1), respectively. MMAs$^V$ is only detected in two precipitation events (P6 and P9.2), while other identified As species are detected in all collected precipitation samples. Concentrations of As species in cloud water are generally higher than in precipitation samples, most likely due to differences in liquid water content (i.e. dilution vs. concentration effect)[33]. Species concentrations in cloud water range from 0.01 to 0.35 $pg \cdot m^{-3}$ for MMAs$^V$ (median of all sub-events: 0.05 $pg \cdot m^{-3}$), from 0.02 to 0.50 $pg \cdot m^{-3}$ for DMAs$^V$ (median: 0.16 $pg \cdot m^{-3}$), from 0.02 to 3.26 $pg \cdot m^{-3}$ for TMAs$^V$O (median: 0.95 $pg \cdot m^{-3}$) and from 0.1 to 7.7 $pg \cdot m^{-3}$ for iAs (median: 2 $pg \cdot m^{-3}$; Fig. 1b), respectively. These concentrations demonstrate that substantial amounts of As are present in its methylated forms, with proportions up to 43% in precipitation (average for all methylated As species: 28 ± 10% of total sum As

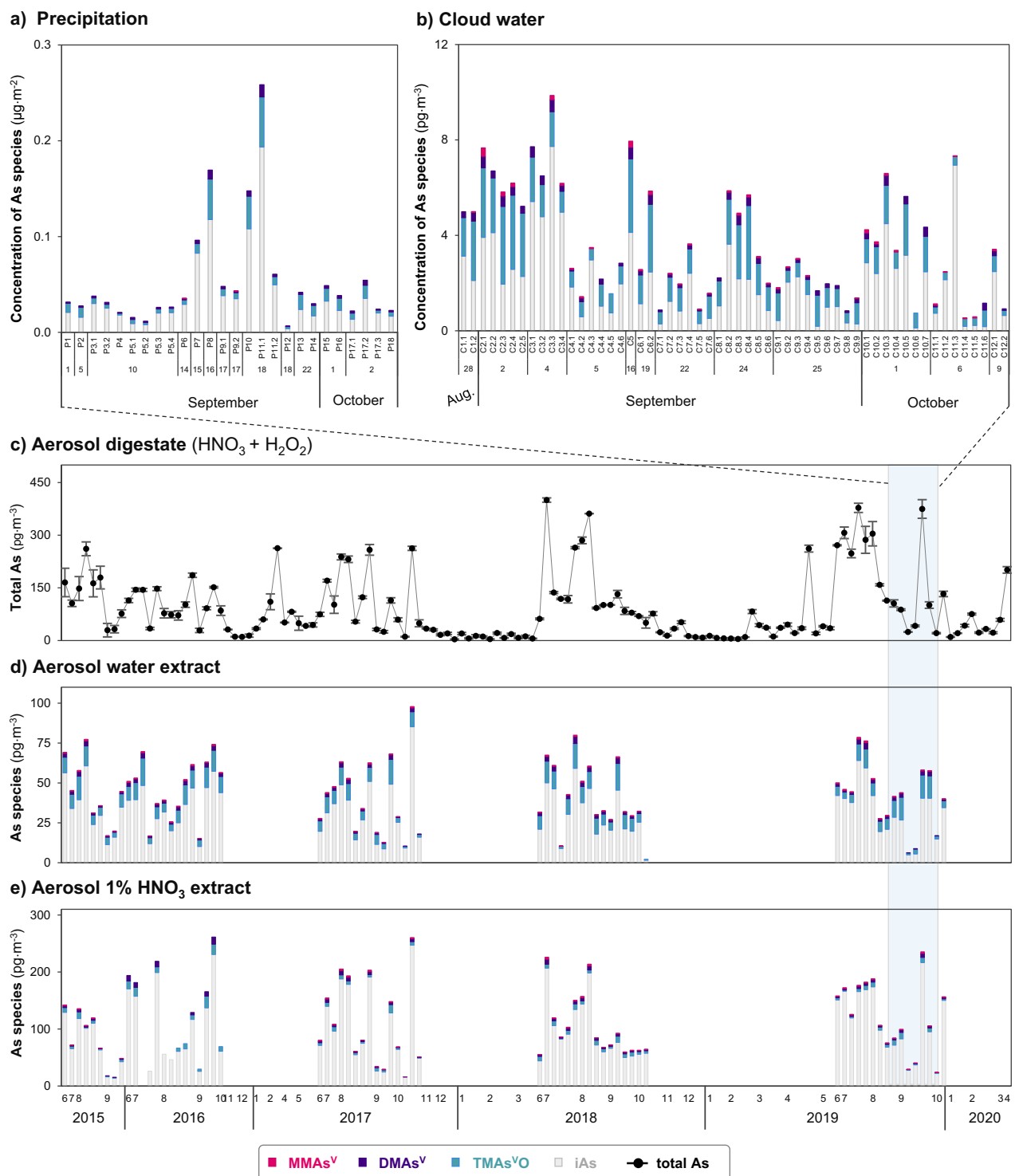

**Fig. 1 | Variability of arsenic (As) species concentrations in precipitation, cloud water samples and different aerosol extracts. a** and **b** show the concentrations of As species in respectively, precipitation and cloud water (sub)events collected during the 2-month campaign in 2019. **c** shows the total As concentrations in the aerosols time series (2015–2020) determined by ICP-MS/MS after acid digestion (using concentrated $HNO_3$ and $H_2O_2$), while **d** and **e** show the concentrations of As species in aerosol water and 1% $HNO_3$ extracts, respectively. The error bars represent the standard deviation values resulting from quantification in triplicate. The x-axis shows the sampling dates, and for **c**–**e**, the 2019 campaign period is highlighted in blue. The shown As species in **a**, **b**, **d** and **e** include monomethylarsonic acid ($MMAs^V$, pink), dimethylarsinic acid ($DMAs^V$, purple), trimethylarsine oxide ($TMAs^VO$, cyan), and inorganic As (iAs; grey).

species) and up to 89% in cloud water (average: $46 \pm 20\%$; Fig. 2), respectively.

Overall, our results indicate a high variability in the proportion of inorganic and methylated As species in wet deposition samples. While the proportions of $DMAs^V$ and $TMAs^VO$ negatively correlate to iAs in both precipitation and cloud waters ($p < 0.01$; Supplementary Fig. 4a, b), the proportions of $DMAs^V$ and $TMAs^VO$ positively correlate with each other ($p < 0.05$; Supplementary Fig. 4a, b). These relationships indicate that inorganic and methylated As species have different origins and/or are affected by different atmospheric processes in wet deposition.

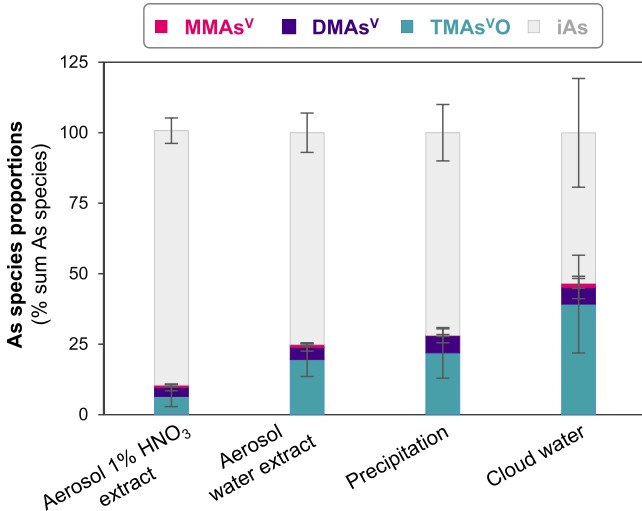

**Fig. 2 | Variability of proportions of arsenic (As) species in different atmospheric deposition samples.** Species proportions (% sum of As species concentrations in respective extracts or sample types) of monomethylarsonic acid (MMAs$^V$, pink), dimethylarsinic acid (DMAs$^V$, purple), trimethylarsine oxide (TMAs$^V$O, cyan) and inorganic As (iAs, grey) in the 1% HNO$_3$ aerosol extract, the aerosol water extract, as well as in precipitation and cloud water.

## High solubility of methylated As species in aerosols

Total As concentrations in aerosols (following acid digestion with HNO$_3$ and H$_2$O$_2$) collected at weekly resolution over 5 years (2015-2020) range from 0.004 to 0.400 ng·m$^{-3}$ ($n = 134$; average 0.092 ± 0.094 ng·m$^{-3}$, Fig. 1c). Similarly to wet deposition samples, these concentrations are considerably lower compared to aerosols collected in semi-rural areas (0.08−37.4 ng·m$^{-3}$)[22,24,34] and urban environments in Austria (0.06−3.3 ng·m$^{-3}$)[23] and China (1−34 ng·m$^{-3}$)[35,36].

In the aerosol samples taken during summer and autumn ($n = 70$), we quantified As species in two types of extracts (Fig. 1d, e), namely (i) ultrapure water extract (with sonication), and (ii) 1% HNO$_3$ extract (microwave-assisted extraction, a conventional approach to extract As species in rice[37]). The sum of As species in the ultrapure water extract is on average 38 ± 17% of total As, while the sum of As species in the 1% HNO$_3$ extract accounts for 87 ± 33% of total As. These extraction efficiencies are similar to previous studies that used diluted H$_2$O$_2$[18,22–24] or hydroxylammonium chloride[34] for extraction (42-89%; Supplementary Discussion 1).

Average concentrations of MMAs$^V$, DMAs$^V$, TMAs$^V$O and iAs in the analysed 2015−2020 aerosols account for 0.5 ± 0.3 pg·m$^{-3}$, 2 ± 1 pg·m$^{-3}$, 8 ± 4 pg·m$^{-3}$ and 32 ± 17 pg·m$^{-3}$ in the water extracts (Figs. 1d) and 1 ± 0.8 pg·m$^{-3}$, 4 ± 3 pg·m$^{-3}$, 6 ± 4 pg·m$^{-3}$, and 98 ± 61 pg·m$^{-3}$ in the 1% HNO$_3$ extracts (Fig. 1e), respectively. Relatively more methylated As species are present in the water extract (25 ± 7% of total sum As species) than in the 1% HNO$_3$ extract (10 ± 5%), suggesting that these species are more soluble than inorganic As. This result is consistent with the substantial proportions of methylated As observed in precipitation and cloud water samples (28 ± 10% and 46 ± 20%, respectively; Fig. 2). A higher proportion of methylated As in water extracts is not only found for the full aerosol series but also in the time period matching the high-resolution precipitation and cloud water sampling campaign (i.e. end of August - beginning of October; Supplementary Fig. 5). Inorganic As is the dominant species in both types of extracts (water extract and 1% HNO$_3$). However, the water extracts have lower proportions of inorganic As than the 1% HNO$_3$ extracts, which is likely related to an insoluble recalcitrant As fraction (in water) potentially containing As-mineral and/or organic colloidal complexes (as commonly observed in soils[38] and surface waters[39]). Future studies are

needed to better characterize the recalcitrant As fraction in aerosols. However, this would require the development of specific extraction methods and/or the use of solid phase speciation techniques, both of which demand high element contents and are therefore not easily applicable to aerosols collected in remote areas and/or with high temporal resolution.

Thus, the combined analyses of aerosol, cloud water, and precipitation samples indicate that methylated As species in aerosols are more soluble than inorganic As. This higher solubility may partly explain the relatively high wet deposition fluxes of methylated As species, thereby impacting the atmospheric supply of As species to surface environments. As for the wet deposition samples, the proportions of DMAs$^V$ and TMAs$^V$O (as % total As in the aerosol extract) negatively correlate to iAs ($p < 0.01$; Supplementary Fig. 4c), and positively correlate with each other ($p < 0.01$; Supplementary Fig. 4c).

## Identification of contributing moisture sources and source indicators

To investigate the origin of methylated As species in atmospheric deposition, we determined the contributing moisture sources as well as different chemical proxies of atmospheric sources, and then analysed, in following sections, the relationships between these source indicators and the different As species.

For the precipitation samples ($n = 26$), the (sub)event sampling approach enables us to capture more distinct contributing moisture source regions and atmospheric transport patterns than for lower resolution sampling approaches (e.g. with a weekly frequency)[40]. At the beginning and end of the 2-month campaign, precipitation (sub)events are characterized by high contributions from distant sources (distances > 500 km) coming from the North Atlantic (average moisture contribution of 58 ± 17% in P2-5 and P13-18 versus 8 ± 7% in other events; Fig. 3a). In the middle period of the campaign, precipitation (sub)events have higher contributions from North Africa (average moisture contribution of 26 ± 9% in P6-P10 versus 1 ± 1% in other events). To identify groups of precipitation samples with significant (dis)similarities in contributing moisture source regions, we applied hierarchical cluster analysis to the moisture source dataset (Fig. 3b). Averages and standard deviations of moisture sources of individual precipitation sub-events within each of the 4 identified clusters are given in Supplementary Table 3. Both clusters 1 and 3 are dominated by moisture uptakes over land (70−72% of total moisture sources), but cluster 1 (number of precipitation (sub)events: $n = 4$) is mainly influenced by moisture coming from Spain (40 ± 5%) and France (16 ± 4%), while cluster 3 ($n = 6$) by moisture originating from North Africa (26 ± 9%) and the Mediterranean Sea (23 ± 3%). Compared to clusters 1 and 3, clusters 2 and 4 show higher contributions from the Atlantic (46−47%), but along different pathways. In cluster 2 ($n = 12$), the moisture follows a direct trajectory from the North Atlantic (62 ± 19%) via the Bay of Biscay (15 ± 5%) to the Pic du Midi sampling site in the French Pyrenees, whereas in cluster 4 ($n = 4$), Atlantic moisture (46 ± 5%) travels over Europe with uptakes over Spain (27 ± 2%) before reaching Pic du Midi. Furthermore, among the clusters, cluster 4 shows the highest contributions from coastal regions (Fig. 3b).

To support the interpretation of each precipitation cluster in terms of dominant source regions, we use different chemical proxies analysed in precipitation. We find significantly higher methane sulfonic acid (MSA) proportions (given in % total S species) in cluster 2, and to a lesser extent in cluster 4 (Fig. 3c), than in clusters 1 and 3, which independently indicates an important marine source as MSA is a well-known proxy for marine DMS emissions[41]. Data for dimethyl sulfone: DMSO$_2$, another marine S proxy, is shown in Supplementary Fig. 6. We also looked at Na content, an indicator of sea spray[42,43], which we normalized by strontium (Sr; Fig. 3d). In a previous study, we showed significantly higher wet deposition fluxes for most analysed (trace) elements in the period 14th−19th September of the campaign

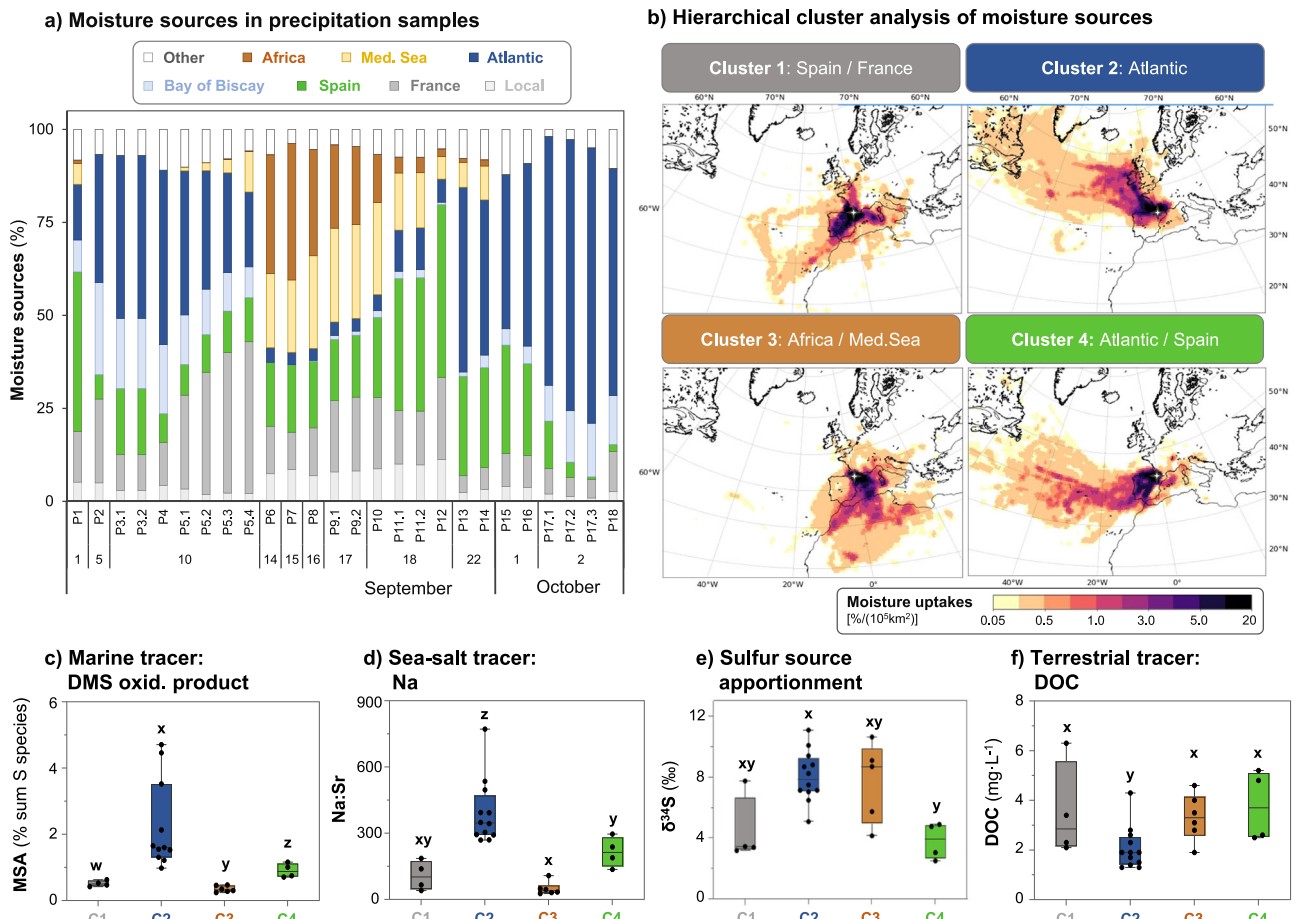

**Fig. 3 | Moisture source contributions and chemical source proxies in precipitation.** **a** shows the temporal variability of moisture sources in precipitation (sub)events collected during the 2019 campaign, previously published in Breuninger et al.[44]. **b** shows the output of the hierarchical cluster analysis on contributing moisture sources for 4 main groups of precipitation (sub)events with predominant moisture sources from Spain and France (cluster 1 -C1-, grey), North Atlantic (cluster 2 -C2-, blue), North Africa and the Mediterranean (cluster 3 -C3-, brown), and North Atlantic Spain and Portugal (cluster 4 -C4-, green). Each cluster is shown with average moisture contributions to the sampling site Pic du Midi Observatory (indicated by white cross). Moisture uptakes are displayed as percentage uptake per area (shown in colour scale 0–20% per $10^5$ km²), thus depending on the defined size of areas, high moisture uptakes (dark purple) do not necessarily correspond to predominant sources. Specific contributions of moisture uptakes in pre-defined regions within identified clusters are listed in Supplementary Table 3. Moisture source plots were compiled with the open-source software Python using the matplotlib[77] and cartopy modules. **c**–**f** show the variability of different source proxies in the four clusters (C1-C4), i.e., methane sulfonic acid (MSA) proportions (% sum sulfur (S) species, **c**), the sodium to strontium (Na:Sr) concentration ratio (sea salt proxy; **d**), the isotopic composition of sulfur ($\delta^{34}S$, **e**), and dissolved organic carbon concentrations (DOC, **f**). The boxplots in **c**–**e** show the interquartile range, representing the middle 50% of the data, which fall between the upper quartile (75% data below that score) and the lower quartile (less than 25% below that score). The whiskers refer to the 5th/95th percentiles. Different letters (x, y, z) in **c**–**e** denote significance levels based on Mann-Whitney-U test ($p < 0.05$), i.e. for all clusters with the same letter, the difference is not statistically significant.

(Supplementary Fig. 7), which was associated with a particular meteorological situation influenced by elevated dust levels as well as thunderstorms[44]. The normalization of Na by Sr content aims at removing the influence of this period and focus more on background conditions rather than specific events with unusually high wet deposition; Sr concentrations are most suitable as Sr is the only element that showed high elemental deposition exclusively during this particular meteorological period (Supplementary Fig. 7). Also, the Na:Sr proxy confirms that cluster 2, and to a lesser extent cluster 4, are importantly influenced by marine sources given the significantly higher Na:Sr ratios than for clusters 1 and 3 (Fig. 3d).

The isotopic signatures of S ($\delta^{34}S$) further support the identified moisture source clusters (Fig. 3e). Cluster 1 and 4 have relatively depleted $\delta^{34}S$ values (i.e. on average +4.8 ± 2.6‰ and +3.8 ± 1.2‰, respectively), which are consistent with previously reported isotopic signatures of terrestrial sources associated with anthropogenic sulfate ($\delta^{34}S$, ~0 to +5‰)[45,46]. For cluster 4, the depleted $\delta^{34}S$ values, reflecting terrestrial sources, combined with marine sources (from marine

source proxies) agree with mixed marine and land contributions as indicated by moisture source analysis. Cluster 2 shows the most enriched $\delta^{34}S$ values, ranging from +5.1 to +11.1‰ (average $\delta^{34}S$ = +8.1 ± 1.7‰), likely related to influences from marine DMS emissions ($\delta^{34}S$ ~ +15 to +21‰)[45,46] in addition to terrestrial sources. Consistent with predominant African moisture sources, $\delta^{34}S$ values in cluster 3 range from +4.1 to +10.6‰ (average $\delta^{34}S$ = +7.7 ± 2.6‰), reflecting previously reported isotopic signatures of mineral dust ($\delta^{34}S$ = +7.4 ± 3.1‰)[46]. Finally, dissolved organic carbon (DOC), which is known to be associated with a range of terrestrial biogenic and anthropogenic sources[47,48], is significantly lower in cluster 2, dominated by marine moisture, compared to cluster 1, 3, and 4 that have higher contributions of moisture from land and/or coastal regions (Fig. 3f).

While we did not use the aerosol samples for back trajectory analyses due the weekly time resolution that would lead to overlapping source contributions, we did analyse them to determine main groups of organic compounds as a further source proxy. It should be noted

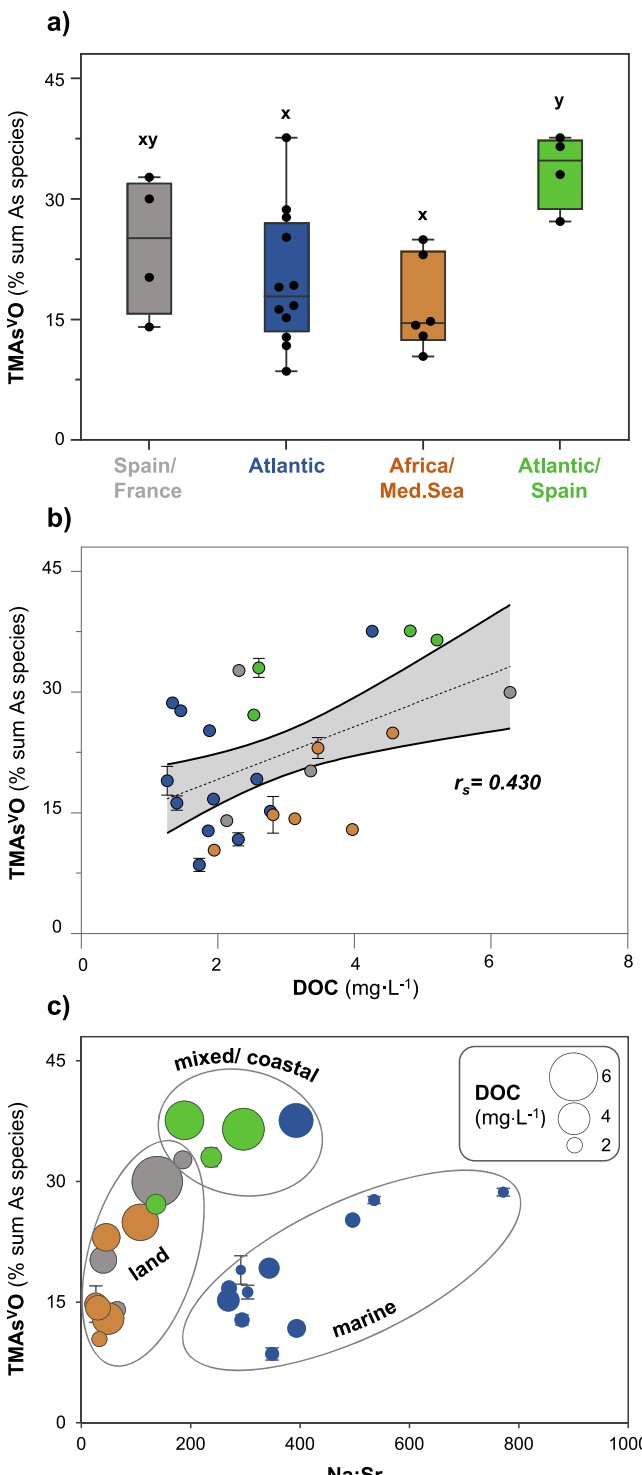

**Fig. 4 | Relationships between the proportions of trimethylarsine oxide (TMAsᵛO) in precipitation (sub)events and contributing moisture sources, dissolved organic carbon content and the sea salt proxy. a** shows the variability of TMAsᵛO proportions (% sum of arsenic (As) species concentrations) in the clusters C1-C4 defined by different contributing moisture sources in Fig. 3b. The boxplots show the interquartile range, representing the middle 50% of the data, which fall between the upper quartile (75% data below that score) and the lower quartile (less than 25% below that score). The whiskers refer to the 5th/95th percentiles. **b** and **c** show the variability of TMAsᵛO proportions as a function of dissolved organic carbon (DOC) and the sodium to strontium concentration ratio (Na:Sr, sea salt proxy), respectively. The size of data points corresponds to the DOC concentrations in precipitation samples. All symbols in **b** and **c** are colour coded according to moisture source clusters shown in Fig. 3b. Different letters (x, y) in **a** denote significance levels based on Mann-Whitney-U test ($p < 0.05$). For all clusters with the same letter, the difference between species proportion is not statistically significant. Indicated correlation coefficient in **b** is significant ($n = 26$, $p < 0.05$), the 90% confidence interval of the regression line (dashed line) is shown in grey, and the error bars represent the uncertainty of quantification by HPLC-ICP-MS/MS performed in duplicate (average relative standard deviation of TMAsᵛO measurements: $4 \pm 3\%$), with invisible error bars indicating standard deviations within the symbol. The ellipses drawn around the points indicate suggested source contributions. The corresponding figure for dimethylarsinic acid (DMAsᵛ) and inorganic As (iAs) is shown in Supplementary Fig. 10.

carboxylic acids (4–42%, Supplementary Fig. 8), which are likely of biogenic origin[44,50,51].

### TMAsᵛO has mixed terrestrial and marine biogenic sources with likely important coastal contributions

Significantly higher proportions of TMAsᵛO are found in cluster 4 (Atlantic/Spain), and to a lesser extent in cluster 1 (Spain/France), in comparison to clusters 2 and 3 (Mann-Whitney-U test, $p < 0.05$; Fig. 4a). Sources originating from Portugal and Spain seem particularly important for TMAsᵛO according to positive correlations between moisture sources from these regions and TMAsᵛO proportions ($n = 26$, $p < 0.05$), which is not observed for the other As species (Supplementary Fig. 9). The mixed marine and terrestrial moisture source contribution and chemical proxy data suggest that both marine and terrestrial contributions are important for TMAsᵛO occurrences. Compared to the more terrestrial clusters 1 and 3, cluster 4 has more marine influences (more MSA and higher Na:Sr ratio; Fig. 3c, d) but compared to the Atlantic cluster (cluster 2), cluster 4 has more terrestrial influences, i.e. positive correlation between TMAsᵛO and DOC ($r_S = 0.439$, $n = 26$, $p < 0.05$, Fig. 4b) and a more terrestrial signature in $\delta^{34}$S (Fig. 3e). Potential terrestrial sources of TMAsᵛO in atmospheric deposition include volatile emissions of trimethylarsine from soils and peatlands[12], volcanoes[16], as well as sewage sludge[52] and biogas plants[16,53] and potential marine sources are emissions by the marine biosphere[13,21,54]. Based on the mixed signal and important contribution of coastal moisture sources, including the northern, western, and south-western coast of the Iberian Peninsula, we suggest that coastal emissions of TMAsᵛO and/or its precursor trimethylarsine could be particularly relevant (Fig. 4c). However, as the number of (sub)events in cluster 4 is low, the mixed source and coastal contributions warrant further investigations.

Cluster 3, which is dominated by moisture sources coming from Africa and the Mediterranean Sea, shows the lowest proportions of TMAsᵛO, providing no evidence for a significant input of volcanic emissions from Mt. Etna (Sicily, Italy) during the 2019 campaign. Finally, in aerosols, we find indications of a dominant biogenic source for TMAsᵛO, as proportions of TMAsᵛO (in both water and 1% HNO₃ extracts) significantly correlate with the abundance of organic compounds from biogenic origin (plants, bacteria and/or algae), i.e. carbohydrates (detected as (cyclo)pentenone/-dione and (alkyl-)furan/furanone by Py-GC/MS)[55,56] and N compounds (Supplementary

that these analyses were done on the whole sample (i.e. total pyrolysis of aerosol filter) and not on liquid extracts as were used for As speciation analyses. Organic compound groups were identified by pyrolysis-GC/MS, a high throughput method that only requires little carbon mass[49,50]. As in the few previous studies that used this technique for atmospheric samples, we identify n-alkanes, n-alkenes, (poly) aromatics, and phenols that have been previously linked to anthropogenic sources and urban environments[44,50,51]. These organic compound groups account for 6–30, 10–24, 2–12 and 2–7% of the sum of peak area of identified groups in analyzed summer and autumn aerosols, respectively (Supplementary Fig. 8). We also find large proportions of carbohydrates (3–26%) nitrogen (N) compounds (7–46%), and

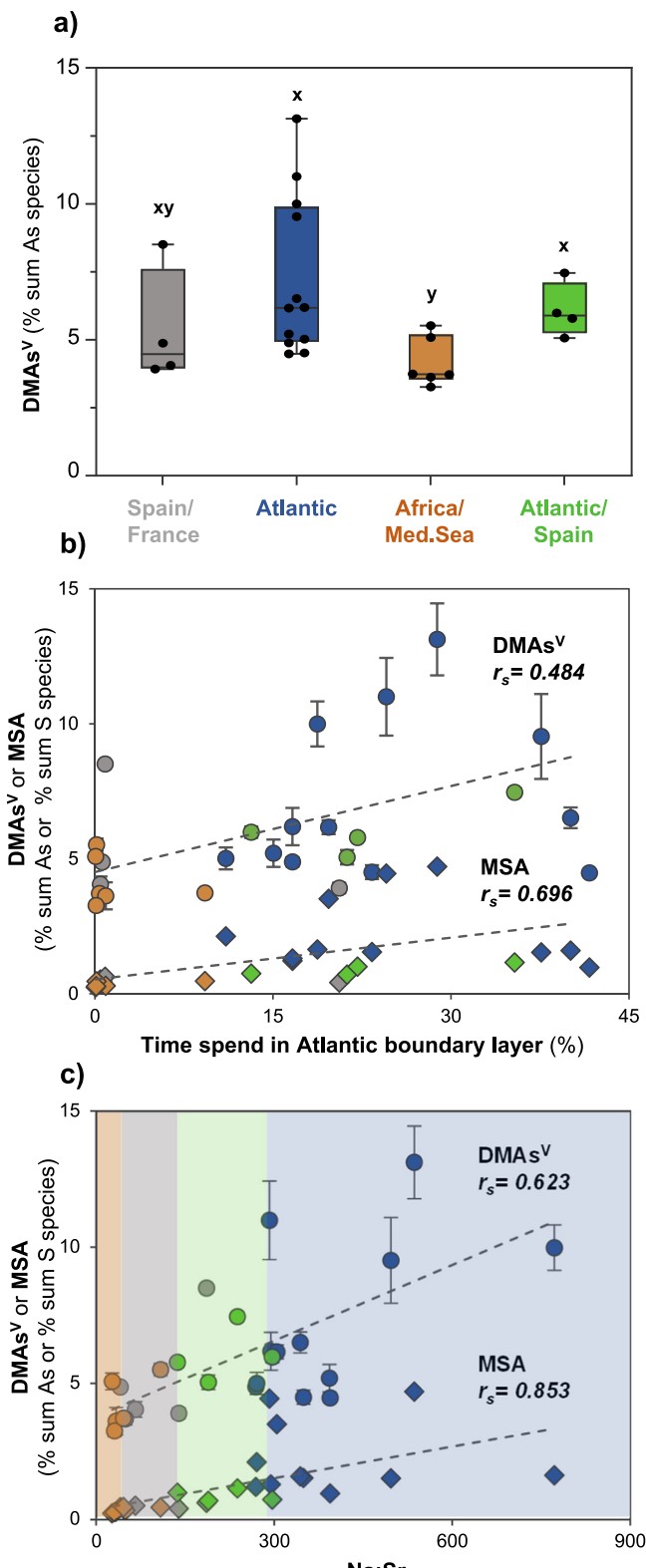

**Fig. 5 | Relationships between the proportions of dimethylarsinic acid (DMAs$^V$) in precipitation (sub)events and contributing moisture sources, time spent in the Atlantic boundary layer, and the sea salt proxy. a** shows the variability of DMAs$^V$ proportions (% sum of arsenic (As) species concentrations) in the clusters C1-C4 defined by different contributing moisture sources in Fig.3b. The boxplots show the interquartile range, representing the middle 50% of the data, which fall between the upper quartile (75% data below that score) and the lower quartile (less than 25% below that score). The whiskers refer to the 5th/95th percentiles. **b** and **c** show the variability of DMAs$^V$ (circles) and methane sulfonic acid (MSA, diamonds) as a function of the time spent in the Atlantic boundary layer (**b**), and as function of the sodium to strontium concentration ratio (Na:Sr, sea salt proxy, **c**). All symbols in **b** and **c** are colour coded according to moisture source clusters. Different letters (x, y) in **a** denote significance levels based on Mann-Whitney-U test ($p < 0.05$). For all clusters with the same letter, the difference between species proportion is not statistically significant. Indicated correlation coefficient in **b** and **c** are significant ($n = 26$, $p < 0.01$) and the error bars represent the uncertainty of quantification by HPLC-ICP-MS/MS performed in duplicate (relative standard deviation of all DMAs$^V$ measurements: $6 ± 4$%), with invisible error bars indicating standard deviations within the symbol. The corresponding figure for trimethylarsine oxide (TMAs$^V$O) and inorganic As (iAs) is shown in Supplementary Fig. 14.

proxies of DMS emissions, i.e. MSA (Fig. 3c) and DMSO$_2$ (Supplementary Fig. 6). Consistent with this, the proportions of DMAs$^V$ positively correlate with the mean time spent in the Atlantic boundary layer ($r_S = 0.484$, $n = 26$, $p < 0.01$; Fig. 5b) and the proportions of MSA ($r_S = 0.538$, $n = 26$, $p < 0.01$; Supplementary Fig. 15b). The proportions of DMAs$^V$ also negatively correlate with the mean time spent in the continental boundary layer ($r_S = -0.487$, $n = 26$, $p < 0.05$; Supplementary Fig. 15a). These correlations are not observed for the other As species (Supplementary Figs. 14, 15). Moreover, the proportions of DMAs$^V$ ($r_S = 0.623$, $n = 26$, $p < 0.01$) and MSA ($r_S = 0.853$, $n = 25$, $p < 0.01$) significantly correlate with the Na:Sr ratio, an indicator of sea-spray (Fig. 5c), reflecting the association of these species with clusters influenced by North Atlantic moisture (clusters 2 and 4). Previous investigations of North Atlantic seawater reported DMAs$^V$ as the main organic As species[54]. Its formation was proposed as a potential detoxification mechanism, i.e. after uptake of As$^V$ by marine algae and reduction, As$^{III}$ is methylated to the less toxic species DMAs$^V$ within the cell, which can then be excreted into the surrounding seawater[14].

A principal component analysis performed on the precipitation dataset including moisture sources, As species, DOC, S species and S isotope composition, confirms the stronger relationship of DMAs$^V$ with indicators for marine sources, in comparison to TMAs$^V$O, which has a closer relationship with proxies of terrestrial sources, again indicating its mixed and potential coastal origin, see Supplementary Fig. 11a, b (Detailed description of principal component analysis and results in Supplementary Note 1). Finally, the proportions of DMAs$^V$ in the water extracts of the aerosol time series positively correlate with those of MSA ($r_S = 0.472$, $n = 70$, $p < 0.01$), in contrast to the other As species (Supplementary Fig. 16), supporting a dominant marine source. Similar to TMAs$^V$O, DMAs$^V$ correlates with carbohydrates (Supplementary Table 4), again indicating a biogenic source.

## No indications for an atmospheric abiotic source of methylated As species

Besides primary emissions sources, it has been hypothesized that methylated As could originate from abiotic As methylation in the atmosphere[58]. In lab experiments, Guo et al.[58] demonstrated the formation of volatile (methylated) arsines by photo-transformation of inorganic As (As$^{III}$) in the presence of low molecular weight organic acids under UV-C irradiation (100–280 nm). However, UV-C irradiation is largely absorbed and/or reflected in the stratospheric ozone layer, and therefore UV-B irradiation is more environmentally relevant. We therefore tested the relevance of abiotic formation of methylated As species under conditions relevant for atmospheric samples by

Table 4)[57]. However, it should be noted that Py-GC/MS analysis cannot distinguish between terrestrial, coastal or marine biogenic sources, which could be addressed in follow-up studies.

## DMAs$^V$ has a dominant marine origin

The highest proportions of DMAs$^V$ in precipitation samples are observed when Atlantic moisture sources are dominant (clusters 2 and 4; Mann-Whitney-U test, $p < 0.05$, Fig. 5a), along with the highest levels of marine

irradiating mixtures of As$^{III}$, organic acids (formic, acetic and oxalic acid), and dissolved organic matter (as photosensitizer) using environmentally relevant UV irradiation (i.e. 290–345 nm, Supplementary Discussion 3 and Supplementary Figs. 12, 13). Under all tested experimental conditions, As$^{III}$ is quantitatively converted to As$^V$, with no detectable amounts of methylated As species for up to 6 h irradiation in ultrapure water and rainwater collected at Pic du Midi, suggesting no considerable abiotic contribution of methylated As to our samples.

## Inorganic As and MMAs$^V$ are linked to terrestrial (anthropogenic/biogenic) sources

The proportions of inorganic As, which is the most abundant form of As, did not vary between clusters of precipitation (sub)events characterized by different moisture sources, except for cluster 4, which has relatively less inorganic As than other clusters (Atlantic/Spain; Supplementary Fig. 10l). The small variations in different clusters could be related to the fact that inorganic As is the final oxidation product of various emitted volatile As species from both anthropogenic and biogenic processes (e.g. As$^{III}_2$O$_3$, As$^{III}$H$_3$). In the aerosol series, inorganic As proportions positively correlate with aliphatic compounds, alkanones and phenols ($p < 0.01$; Supplementary Table 4) identified by Py-GC/MS, which have been linked to various terrestrial sources including coal combustion and urban anthropogenic emissions[51].

MMAs$^V$ is only detected in two precipitation events and therefore not included in the cluster analysis. However, in aerosols, MMAs$^V$ is detected in 94% of samples and does not indicate any link to biogenic emissions sources (i.e. degradation products of carbohydrates and proteins) as observed for other methylated As species. Rather, it shows a terrestrial, likely anthropogenic source signature as indicated by positive correlations to aliphatic compounds (alkanes) and phenols (Supplementary Table 4). Furthermore, a significant correlation between MMAs$^V$ proportions in aerosols and moistures sources from Western Europe (i.e. over Spain and surrounding regions within a ~ 0.5° radius of Pic du Midi) suggests a relatively local and/or regional source signature for MMAs$^V$, which may also explain the link to anthropogenic sources. Indeed, analysis of a few aerosol samples collected during winter (most winter samples contained insufficient material for analysis) shows relatively high proportions of MMAs$^V$ (Supplementary Fig. 17), potentially indicating regional contributions from biomass burning, which takes place in February-March.

## Deposition flux of methylated As to the surface

The fate of As in terrestrial surface environments is strongly influenced by its chemical speciation, as previously shown for irrigation water[59,60]. For example, As$^{III}$ is highly mobile in soils and available for plant uptake, while As$^V$ is efficiently retained by sorption to oxide minerals and thus less available to plants under oxic conditions[61]. MMAs$^V$ and DMAs$^V$ have been commonly detected in soils[62], and to a lesser extent TMAs$^V$O[63]. DMAs$^V$ has a higher accumulation rate in rice plants than inorganic As species due to its more efficient transfer from roots to grains[25], resulting in an increased risk of dietary exposure. To investigate the potential impact of atmospheric inputs of methylated As to soil systems, we compare estimated deposition fluxes of methylated As species based on our data with previously reported As methylation rates in soils.

Based on our precipitation data collected during the high-resolution campaign in 2019, we calculate a deposition flux of methylated As species (sum of TMAs$^V$O, DMAs$^V$, MMAs$^V$) between 0.005 and 3.38 µg·L$^{-1}$·d$^{-1}$ (average: $0.5 \pm 0.8$ µg·L$^{-1}$·d$^{-1}$ considering rain rates during the campaign; Fig. 6; Supplementary Discussion 4). Using previously published measurements of methylated As species determined in pore water of natural soils (pot experiments with paddy and arable soils from Bangladesh, China, UK and USA) with varying contamination levels (contamination by irrigation, mining, high geogenic background or no contamination)[62,64], we estimate an in-soil production rate between 0.002 and 0.41 µg·L$^{-1}$·d$^{-1}$ (average:

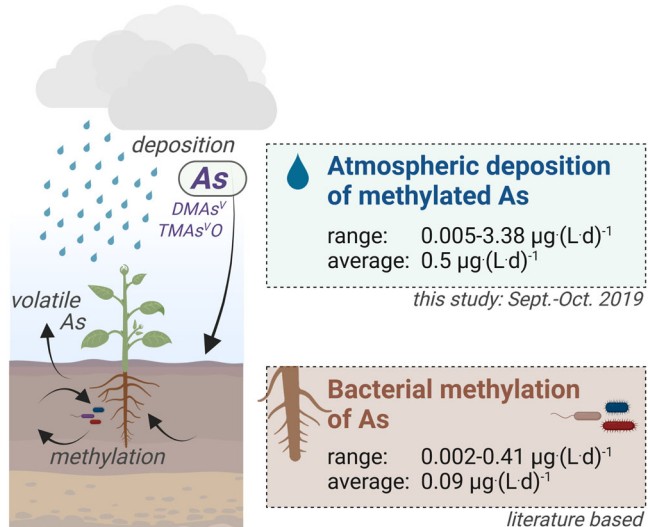

**Fig. 6 | Comparison of atmospheric deposition flux of methylated arsenic (As) species and previously reported bacterial methylation rates of As in soils.** DMAs$^V$ refers to dimethylarsinic acid and TMAs$^V$O refers to trimethylarsine oxide. Atmospheric deposition fluxes are based on precipitation data of sub-events considering rain rates during the campaign. Bacterial methylation rates of As in soils are based on published data[62,64]. Created in BioRender. Breuninger, E. (2023) BioRender.com/x59g039.

$0.09 \pm 0.15$ µg·L$^{-1}$·d$^{-1}$ considering methylation efficiencies per incubation time; Fig. 6; Supplementary Discussion 4 and Supplementary Table 5), which is lower than the average wet atmospheric inputs we estimate.

Although the extent of atmospheric deposition of methylated As species will vary locally depending on deposition patterns as well as contributing emissions sources, our findings demonstrate the potential quantitative importance of atmospheric inputs of methylated As to soil systems. It is worth pointing out that biogenically-sourced TMAs$^V$O and DMAs$^V$ present in atmospheric deposition, may include so-called legacy As similar to mercury[10]. Anthropogenic emissions have decreased in the last decades, particularly in the Northern Hemisphere, but the identified biogenic methylated As species at Pic du Midi may still originate from anthropogenically derived As from various sources (e.g. mining, coal burning, metal smelting, wood preservatives, herbicides), that have been re-emitted via biogenic processes[65].

Our approach of combining speciation techniques with other chemical proxies and moisture source diagnostics offers a unique tool for studying source contributions of individual methylated As species. Our data indicate mixed marine and terrestrial biogenic sources for TMAs$^V$O, including likely coastal contributions, while for DMAs$^V$, we find more distinct marine source signatures (Fig. 7). For both inorganic As and MMAs$^V$, we find indications for terrestrial, likely anthropogenic sources. Further field campaigns are needed to confirm these species-specific source contributions with larger numbers of precipitation (sub)events and/or aerosol samples collected at high temporal resolution. Furthermore, the extent to which the source contributions of methylated As species vary with sampling locations also warrants further investigation. Nevertheless, our results suggest that methylated As species underwent long-range transport and should thus be included in atmospheric budget estimates. In addition, identified source signatures of different As species in this study may be used to prioritize future field studies. For example, investigating (volatile) methylated As species emission fluxes from marine environments by sea spray or via sea-air exchanges. Combined field measurements, as well as experimental and modelling approaches, will be essential to

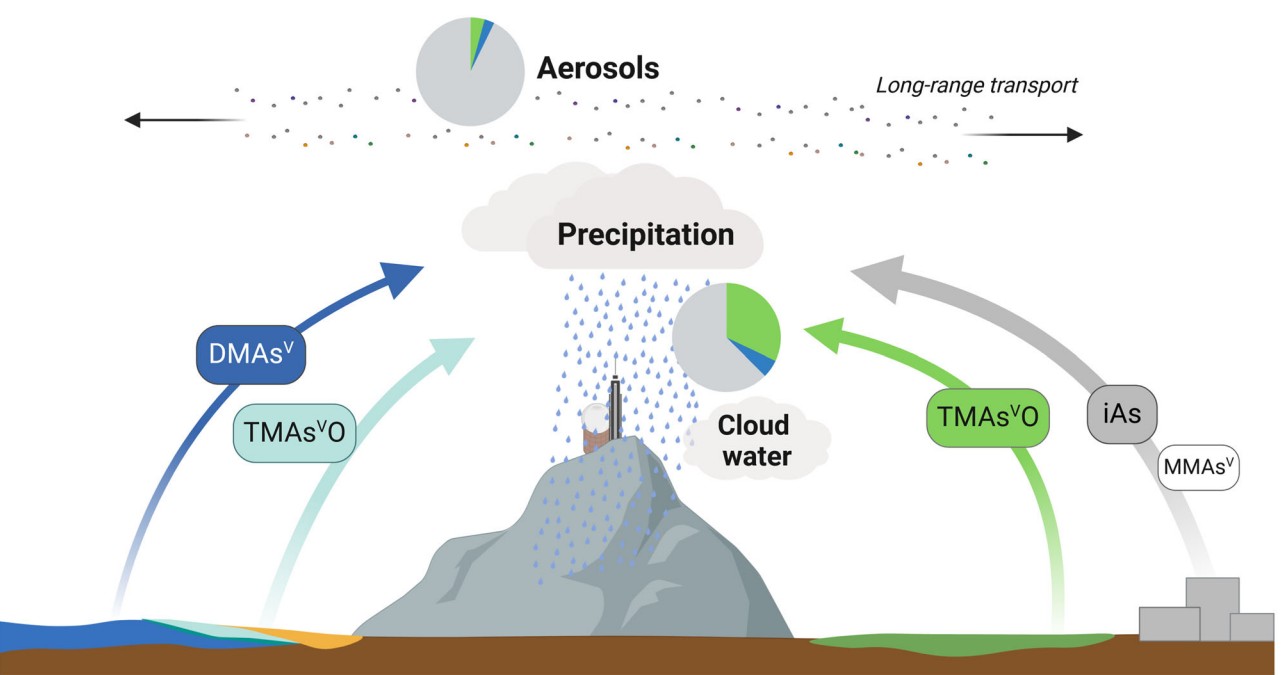

**Fig. 7 | Overview of source contributions for atmospheric arsenic (As) species in deposition samples collected at Pic du Midi Observatory.** DMAs$^V$ refers to dimethylarsinic acid, TMAs$^V$O refers to trimethylarsine oxide, MMAs$^V$ refers to monomethylarsonic acid, and iAs refers to inorganic arsenic. Created in BioRender. Breuninger, E. (2024) BioRender.com/h88t536.

evaluate biogenic contributions to the biogeochemical As cycle and its consequences for ecosystem and human health.

## Methods

### Sample collection

Atmospheric deposition samples were collected at Pic du Midi Observatory (2877 m a.s.l.), which is located in the central Pyrenees Mountains. We collected two series of atmospheric samples at different temporal resolutions: (1) precipitation and cloud water were collected at high temporal resolution (i.e. on a sub-event basis) between 28th August and 13th October 2019; (2) aerosol samples were collected weekly between 23rd June 2015 and 24th April 2020.

**Precipitation and cloud water sampling.** Precipitation samples were collected with three custom-built, pre-cleaned (with 10% HNO$_3$ followed by ultrapure water) polypropylene collectors, which were only opened during precipitation events (surface area per collector: 0.159 m$^2$). Cloud water samples were collected with a CASCC2 sampler (air volume flow: 1470 m$^3$·h$^{-1}$). All plastic storage containers of wet deposition samples were pre-cleaned with 1% HNO$_3$ and then rinsed with ultrapure water. A total of 26 precipitation and 58 cloud water sub-events were collected. The total deposition amounts of collected samples were determined by weight. Sub-samples for total element quantification were filtered (0.22 µm syringe filters, cellulose acetate, BGB), acidified (1% v/v HNO$_3$ of 69%, ROTIPURAN® Supra, Roth) and stored at 4 °C until analysis. Sub-samples used for speciation analyses were filtered (0.22 µm, cellulose acetate), immediately frozen and stored at −20 °C. Sub-samples for dissolved organic carbon (DOC) analysis were filtered (GF/F, 25 mm syringe filters, Whatman®) and stored in pre-burned amber glass vials at −20 °C.

**Aerosol sampling.** Weekly aerosol samples (2015–2020) were collected using two different systems during warm (June-October) and cold (November-May) months. A total of 70 aerosol samples were collected on pre-burned 8 × 10-inch quartz filters (QM-A, Whatman) from June to October between 2015 and 2019 by using a TISCH high-volume PM10 sampler (average collection volume of 7197 ± 1816 m$^3$ over 7 ± 2 days). The weekly sample collection was done with daily sampling during 22:00 to 15:00 UTC between 2015 and 2018, and from 20:00 to 08:00 UTC during 2019. Full-day sampling was not possible due to noise interference of the high-volume sampler with the telescope observations at the station.

Aerosol sampling during colder months required a heated manifold (Tekran 1104 Teflon-coated), which pulls ambient air at a constant flow rate of ~100 L·min$^{-1}$ using a blower unit. Single-stage 90 mm Teflon filter packs (Savillex) equipped with pre-burned quartz filters (QM-A, Whatman) were connected to the manifold. A total of 64 aerosol samples were continuously collected over 6.9 ± 0.4 days with an average collection volume of 518 ± 32 m$^3$ from November to May between 2016 and 2020. One to three filter blanks were taken per collected season. After collection, all aerosol filters were stored in aluminium foil in the dark at −20 °C.

**Pre-concentration of wet deposition samples.** Precipitation sub-samples were pre-concentrated by lyophilization for As and S measurements. The sub-sample volume was reduced from 12 mL to 1.5 mL (pre-concentration factor of 8), then ammonium citrate was added to increase ionic strength (detailed description in Supplementary Discussion 1). Smaller initial volumes of cloud water samples were used depending on sampled amounts.

### Acid digestion of aerosols

For total element analysis, aerosol filters were extracted in 4 mL HNO$_3$ (69%, ROTIPURAN® Supra, Roth) and 1 mL H$_2$O$_2$ (30% (w/w), ultra-trace, Sigma-Aldrich) in a microwave oven (UltraClave IV, MLS). We selected extraction ratios (filter area: digestion volume) of 3.8 cm$^2$:25 mL$^{-1}$ for warm season samples and 7.069 cm$^2$:15 mL$^{-1}$ for cold season samples. We followed the digestion programme of Kulkarni et al., which included a first temperature increase to 180 °C (15 min ramp, temperature held for 15 min) followed by a final increase to 210 °C (15 min ramp, temperature held for 45 min)[66]. The supernatant was filtered (0.22 µm syringe filter, Nylon 66, BGB) and stored at 4 °C until elemental analysis.

## As species extraction in aerosols

We performed two types of extractions to determine i) water-soluble As species and ii) total extractable As species in 1% $HNO_3$ (detailed description in Supplementary Discussion 1). These extractions were solely performed on the warm season samples (collected from June to October 2015–2019) due to limited sample material in the cold season samples.

For the water-soluble fraction, we extracted aerosol filters (11.404 cm$^2$) with 15 mL of ultrapure water. The suspensions were sonicated twice for 20 min at 20 °C, and the resulting extracts were filtered with a 0.22 μm syringe filter (Nylon 66, BGB). 9 mL of the extracts were pre-concentrated by lyophilization analogously to wet deposition samples, while the remaining 6 mL were used to determine extraction efficiencies through total element analysis.

The total extraction of As species included the addition of 4.8 mL 1% (v/v) of 69% $HNO_3$ and 0.2 mL $H_2O_2$ (30%) with an extraction ratio of 14.137 cm$^2$ of filter per 5 mL of liquid. Samples were extracted in a microwave oven (UltraClave IV, MLS) according to the protocol by Sun et al.[67]. The extraction programme included a first temperature increase to 55 °C (5 min ramp, temperature held for 10 min), then to 75 °C (5 min ramp, temperature held for 10 min), and finally to 95 °C (5 min ramp, temperature held for 30 min). After extraction, the suspensions were filtered (0.22 μm syringe filter, Nylon 66, BGB) and stored at 4 °C in acid-cleaned HPLC polypropylene vials until analysis.

## Quantification of total elemental concentrations in wet deposition samples and acid digests of aerosols

As and other (trace) elements were quantified in acidified wet deposition samples and acid digests using an Agilent 8900 ICP-MS/MS. Briefly, the instrument setup consisted of a concentric nebulizer, a Scott double-pass spray chamber cooled to 2 °C, a high-throughput injection system (ISIS) with a PTFE sample loop, and platinum sampler and skimmer cones. As was measured in MS/MS mode with 30 mL·min$^{-1}$ oxygen ($m/z$ 75 → 91) and an acquisition time of 0.3 s. All ICP-MS/MS parameters were optimized using a tuning solution containing 10 μg·L$^{-1}$ of Mg, Li, Tl, Y, Co, and Ce. Instrument drift was monitored and corrected using internal standards (i.e. 1 mg·L$^{-1}$ Sc; 0.1 mg·L$^{-1}$ In, Lu; 50 μg·L$^{-1}$ Y). Instrument performance was checked with two certified reference materials for trace elements in surface waters with dilutions of 1:10 and 1:100 (SRM NIST 1643 f; TMDA 51.2, National Water Research Institute Environment Canada). As yielded recoveries of 99 ± 0.7% (1% $HNO_3$, matrix of wet deposition samples, $n$ = 4) and 103.7 ± 0.7% (16−26% $HNO_3$, acid digests of aerosol samples, $n$ = 6), respectively, for a concentration range of 0.574 to 5.74 μg·L$^{-1}$ (recoveries of other (trace) elements in Supplementary Table 6).

## As and S speciation in pre-concentrated wet deposition samples and aerosol extracts by HPLC-ICP-MS/MS

As speciation was analysed by High Pressure Liquid Chromatography (Agilent 1260 Infinity II Bio-inert HPLC system) coupled to an Agilent 8900 ICP-MS/MS. The chromatographic separation of As species was optimized based on the method published by Raber et al.[68] (detailed description in Supplementary Discussion 2). Briefly, As species were separated by anion exchange chromatography using a PRPX-100 column (Hamilton, 2.1 × 150 mm, 5 μm) equipped with an in-line filter (Titanium Frit 0.5 μm, 10−32 Waters type, BGB), and using isocratic elution (5 mM malonic acid, pH 5.6) with a flow gradient of 0.4−0.5 mL·min$^{-1}$ and an injection volume of 20 μL. Hydrogen peroxide (10% (v/v)) was added prior to the analysis to oxidize any As$^{III}$ to As$^V$ (data given as iAs = sum of As$^{III}$ + As$^V$) and avoid co-elution of As$^{III}$ and TMAs$^V$O. To validate TMAs$^V$O identified by anion exchange chromatography-ICP-MS/MS, a subset of samples were re-analysed by cation exchange chromatography (IonoSpher 5 C, 100 × 3.0 mm) coupled to ICP-MS/MS following the method by Sloth et al.[69], including

gradient elution with 0.5-20 mM pyridine-formate (pyridine anhydrous, 99.8%, Sigma-Aldrich; formic acid ≥ 98%; Sigma-Aldrich) at pH 2.7, with a flow rate of 1 mL·min$^{-1}$ and an injection volume of 25 μL. Sulfur speciation was done according to Müller et al.[70], which includes gradient elution with formic acid (from 24 to 240 mM) delivered at 1 mL·min$^{-1}$ and an injection volume of 50 μL, using a Hypercarb column (100×4.6 mm, 5 μm, Thermofisher) equipped with its guard column.

For As and S speciation analysis, the ICP-MS/MS was operated in MS/MS mode with a mixture of $O_2$ (30%) and $H_2$ (As: 2 mL·min$^{-1}$; S: 1 mL·min$^{-1}$) and an acquisition time of 0.3 s for As ($m/z$ 75 → 91) and 0.05 s for S ($m/z$ 32 → 48; Supplementary Table 7). A solution containing 30 μg·L$^{-1}$ of Y and 14% (v/v) tetramethylammonium hydroxide was continuously supplied post-column by using the peristaltic pump of the ICP-MS/MS. Quantification was done by external species-specific calibration (i.e. As standards prepared in pre-concentrated or aerosol extract matrices). Both As and S speciation analyses were measured in duplicates (i.e. two injections per sample).

All As and S stock solutions were prepared by weight with analytical grade reagents and ultrapure water (18.2 mΩ·cm Thermo Fisher, Barnstead NANOpure DIamond) in acid-cleaned vials and stored in the dark at 4 °C. We used the following standards: arsenite (As$^{III}$, ≥90.0%), arsenate (As$^V$, ≥98%) monomethylarsonic acid (MMAs$^V$), dimethylarsinic acid (DMAs$^V$), trimethylarsine oxide (TMAs$^V$O, 95%) for As measurements; dimethyl sulfoxide (DMSO, ≥99.9%), dimethyl sulfone (DMSO$_2$, ≥99%), methanesulfonic acid (MSA, ≥99.5%), methanesulfinic acid (MSIA, 85%), sodium formaldehyde bisulfite (i.e. hydroxymethanesulfonate: HMS, 95%) and sulfate (SO$_4^{2-}$, 99%) for S measurements. Most stock solutions were purchased from Sigma Aldrich, except for MMAs$^V$ (Chem Service), TMAs$^V$O (MuseChem), SO$_4^{2-}$ and DMSO$_2$ (both purchased from Merck). Working standard solutions were prepared on the day of analysis by dilution in ultrapure water. Limits of detection (LODs) were calculated according to IUPAC recommendations[71], i.e. the LOD equals three times the standard deviation of blank baseline signal divided by the calibration slope based on peak height.

## Quantification of dissolved organic carbon in wet deposition samples

Dissolved organic carbon (filtered with GF/F, Whatman®), specifically the non-purgeable organic carbon (NPOC), was quantified using a total organic carbon analyser (Shimadzu, model TOC-L CSH). The NPOC method used 50 μL injections. The method purge time and gas flow were 1.5 min and 80 mL, respectively. NPOC calibrations were done with recrystallized potassium phthalate (Sigma Aldrich).

## Analysis of organic compounds in aerosol samples by pyrolysis-GC/MS

The analysis of organic compounds in aerosols was carried out using a FrontierLab pyrolyzer (equipped with a FrontierLab AS-1020E autosampler) connected to a GC/MS system (Trace 1310 GC, ISQ 7000 MS, Thermoscientific). Prior to analysis, 1.539 cm$^2$ of aerosol filters were punched and folded into pyrolyzer cups (Eco-cup SF, Frontier Laboratories, Japan). The operating conditions and subsequent data processing method were done according to Tolu et al.[49] (see detailed description in Supplementary Method 1 and Supplementary Table 8).

## Analysis of S isotopes by MC-ICP-MS

S isotopes in precipitation were analysed after offline sulfur isolation by anion exchange chromatography using a multicollector (MC)-ICP-MS (Neptune Plus Thermo Scientific). The detailed protocol is described in Supplementary Method 2 and the MC-ICP-MS operating conditions are listed in Supplementary Table 9. Data were normalized to the Vienna Canyon Diablo Troilite (VCDT) scale using the international reference standard IAEA-S-1.

## Air mass back trajectory and moisture source analysis

Trajectories were calculated using the Lagrangian analysis tool LAGRANTO[72,73] with three-dimensional wind fields from the atmospheric reanalysis dataset ERA5[74] interpolated to a 0.5° × 0.5° horizontal grid on 137 vertical levels. 7-day backward trajectories at 9 vertical levels between 900 and 300 hPa from 5 points at and around Pic du Midi were calculated at 1 h resolution for precipitation and cloud water samples and at 6 h resolution for the 2015–2020 aerosol time series. For the precipitation and cloud water samples, these trajectory datasets were used to identify evaporative moisture source regions (i.e. by tracking specific humidity changes along the trajectory) following the Lagrangian approach developed by Sodemann et al.[75] This moisture source analysis was not performed for the aerosol series.

## Statistical analyses

All statistical analyses were performed using IBM SPSS Statistics 26. Correlations were performed as Spearman rank correlations (correlation coefficient indicated by $r_S$), and significance levels (2-tailed) are indicated by $p < 0.05$ or $p < 0.01$. For the collected precipitation events ($n = 26$), we performed hierarchical agglomerative cluster analysis on the moisture source datasets using Ward's linkages based on Euclidean squared distances to identify their main source regions in terms of dominant contributing moisture sources. Then, for the main variables of interest in this study (e.g. proportions of As species), significant differences between the identified clusters were investigated using the Mann-Whitney U-test with significance levels (2-tailed) indicated by $p < 0.05$ or $p < 0.01$.

## Data availability

The figures and tables of the main manuscript and the supplementary information show all data supporting the findings of this study. Tables with the generated data on arsenic speciation, sulfur isotopes, dissolved organic carbon, as well as modelled moisture sources and time spent in the boundary layer can be found in the Supplementary Dataset 1. All other chemical data can be found on the ETH Zurich Research Collection repository (https://doi.org/10.3929/ethz-b-000647996)[76].

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

## Acknowledgements
We thank the Swiss National Science Foundation (project n°179104: E.S.B, J.T., S.B., L.H.E.W; project n° TMSGI2_218303: F.A.; project n°PP00P2_163661: A.M.), the Swiss Polar Foundation (project DAWATEC: I.T., F.A., E.S.B., J.T., S.B., L.H.E.W.), Eawag and ETH Zurich (E.S.B, J.T., S.B., L.H.E.W, I.T., F.A., R.C., K.M.) as well as the German Research Foundation (DFG, ME 3685/5-1, project n°440953647: D.T., B.M.) for fundings. Samples were collected at the Pyrenean Platform for Observation of the Atmosphere P2OA, and we acknowledge the technical support from the UMS 831 Pic du Midi Observatory team. The authors gratefully acknowledge Heini Wernli (ETH Zurich) for helpful discussions and his feedback on a previous version of the manuscript. The authors thank Stefan Tanda, Walter Goessler, Elke Suess and Patrick Neuhaus for doing preliminary measurements of As speciation on a few previously collected rainwater and aerosol samples. The authors are grateful to their colleagues at Eawag and ETH Zurich, especially Björn Studer for carrying out the dissolved organic carbon analysis, Elyssa Beyrouti for her help in the Py-GC/MS measurements and Caroline Stengel for general lab support.

## Author contributions
E.S.B., J.T. and L.H.E.W. conceptualized the study, with inputs from F.A., I.T. E.S.B. created visualizations and wrote the manuscript with help from J.T., L.H.E.W., and contributions from all the co-authors. J.E.S. conceptualized the observational setup and provided all aerosol samples. E.S.B. and I.T. performed the fieldwork during the 2019 campaign with support from J.T. and J.E.S. E.S.B. carried out sample preparation, chemical analysis and data treatment with support from J.T., S.B. and A.M. J.T. conducted the Py-GC/MS analysis and data treatment. E.S.B. designed photochemical experiments with R.O. and K.M. and E.S.B. performed the experiments. D.T., J.V. and B.M. conducted the S isotope analysis and data treatment. F.A. provided air parcel back-trajectories and moisture sources. E.S.B. conducted all statistical analysis with inputs from J.T. E.S.B. interpreted all findings with help from J.T., L.H.E.W. and with contributions from all other co-authors.

## Competing interests
The authors declare no competing interests.
