## [Transparent Peer Review file · Nature Communications]

Marine and terrestrial contributions to atmospheric deposition fluxes of methylated arsenic species

Corresponding Author: Dr Julie Tolu

Version 0:

Reviewer comments:

Reviewer #1

(Remarks to the Author)

This is a very interesting study which studies for the first time the origin of atmospheric methylated arsenic compounds and quantifies some deposition rates. This is really a missing piece in the puzzle. The foundation of the work is the excellent experimental design to analyse the air mass origin by using different markers in addition to sophisticated arsenic speciation methodology. I have not comment on the analytical methodologies. But before publication the authors should consider the following aspects to improve the manuscript.

Especial aspects:

line 77: Yes the authors are correct that DMA(V) is taken up faster under certain circumstances. But does the exposure to toxic arsenic really increase? The legislation for arsenic in rice considers only inorganic arsenic and so far not methylated arsenic compounds. What is known about TMAO?? This statement should be revised.

Line 147: it is not clear how the errors are calculated. Standard deviations or std errors cannot be used, since 2 ± 3 ng/L makes no sense... Please revise.

Line 154: variable "as in the concentrations of inorganic and methylated As species" is that so, but more important is that the proportion of methylated and inorganic arsenic is variable and that is more important... I think there are not enough events measured that we can assess that terrestrial and marine have different concentrations... but proportions can be discussed...

Line 189: the not quantitative recovery might be similar to previously published data, but that does not make it good. This should further been discussed, since that seems important for the entire calculations (short explanations should be given in the main text especially about the recalcitrant arsenic)

Line 181: "likely including inorganic As-mineral and/or organic colloidal complexes" this is gibberish... I think organically bound arsenic should be solubilized by the acid. I think that is not convincing. As in minerals is more likely. Here data would be good. A pooled sample might be subjected to XRD could give some information, at least if it is amorphous or not. Here more meet on the bone please.

There is an elegant way to determine the origin of the trapped atmospheric samples, but it is not clear how the authors calculate the error used for the classification of the samples with regards to their origin.

Line 292: The authors are correct to say that DMA(V) is the most important methylated species in the seawater. Here more discussion should be done, why this is? Algae metabolism should be described here...

Line 299: Do we expect AsH₃ biogenic or anthropogenic generation ?? There are no volcanoes in Spain? Is this argument really relevant?

Line 347: what is meant by anthropogenic methylated arsenic here?? Unclear do the authors mean herbicide usage or what??

One aspect the reviewer misses. What is the stability of the individual methylated arsenic compounds in the atmosphere? What do we know and would that have a significant influence on the proportion of the individual methylated arsenic compounds? Do we know if de-methylation can take place in the atmosphere in presence of OH radicals? Could that give a bias in the interpretations what the origin of the individual is...

Reviewer #2

(Remarks to the Author)

Summary

1. This is an impressive dataset of methylated arsenic species in troposphere precipitation, based on state-of-the-art analysis of difficult to obtain samples. It should add new insights to a field that is both emerging and of great consequence for arsenic biogeochemical cycling. A cycle of importance due to exposure to humans being of concern at trace concentrations of this element.
2. However, the text over exerts its priority in the field.
3. While precipitation is event orientated the authors then go an pair this with aerosol data which is collected over a week. For modelling event air trajectory origins this pairing is not appropriate.
4. Origin groupings are confused and overlapping, and appear to have low N.
5. The origin data portioning appears imprecise and over-interpreted.
6. Statistics throughout are muddled – see below.
7. Given the above, a key thesis of this report that trimethyl arsine is derived primarily from terrestrial environs cannot be supported.
8. The data analysis needs revisiting, and to be presented in a coherent and robust form so that a new thesis may emerge from that analysis.

Detailed points

1. Regarding priority, line 23 of the abstract states - “However, the speciation of methylated arsenic in deposition and their potential long-range transport is unknown.” Reference 18 in their citation list, was the first to identify methylated arsenic species in precipitation. The long range transport of methylated arsenic species was first explored by reference 17 in the citation list. Other studies have modelled the back trajectories for methylated arsenic species in precipitation, refs 19 and 20 on their reference list. Perhaps the authors mean that they are first to measure methylated species and their long-distant transport in tropospheric samples?
2. The line 23 statement regarding priority is contradicted by line 66 - “While several studies have investigated the occurrence of methylated As species in atmospheric deposition^{17, 18, 19, 20, 21, 22, 23,}”?
3. Line 67 – “the degree to which long- range transport of variable sources can contribute to their deposition is not well described.” This is a confusing statement. Does it mean that the scientific protocols described in previous studies were not clear, or perhaps more likely, that they think their approach is more sophisticated? Please clarify, noting while an incremental improvement in methodology presented here does not invalidate historic studies on which their manuscript builds.
4. Line 69: “Previous studies ... low altitude sites ...” Note also that low altitude and urban sites are perhaps more relevant to actual deposition of arsenic than what is in the troposphere, and thus complimentary to tropospheric studies. It would be good to contrast and compare positives and negatives of both approaches.
5. Line 90: The authors state that aerosol samples are collected weekly – this seems quite out of sync with the resolution of the precipitation data, stated to be “high temporal resolution”, line 85, with which the aerosol data is paired for modelling?
6. Line 136: states: “This analytical advancement ...” Their lyophilization is both obvious and relatively standard. The “advanced” analysis is the arsenic speciation which was developed by others before them. Thus, while their methodology is an incremental, it is useful, improvement with respect to lowering limits of detection.
7. Line 136 states - “...allowed the detection of methylated As species in all collected samples in contrast to most previous studies.” Does this mean that all samples should all samples contain (detectable) methylated arsenic species? They do not detect all methylated species in all samples – see line 142 where MMA was only detected in two samples. What I think the authors are trying to say is that they have improved limits of detection in their arsenic speciation analysis, which is correct, but they have already stated this?
8. Line 136 states - “... most previous studies.” Here “most” is vague. It would be good to know which studies reached the standard of theirs?
9. Line 201 states that 26 events had their sources identified, with only 4 of these from Spain. “N” here is important as the key conclusion of this report is that samples from this region are elevated in percentage TMA₂O compared to marine regions?.
10. Line 226 - paragraph starting at - “TMA₂O has a dominant terrestrial biogenic origin” does not discuss that the only study reporting arsines generation from seawater, reference 13 on their list, found that TMA was, by far, the dominant species produced. This seems somewhat pertinent do the discussion as it is in opposition to their manuscripts primary finding?
11. Line 230 – Spain and Portugal and noted as a source region, but Portugal then disappears from the text? The text seems to indicate that the mountainous regions of N. Spain and S. France are important, not the whole of Spain – but the text and presentation is confused.
12. Figure 4a - the clusters all overlap, all with their epicentre in N.W. Spain, yet much is made of these cluster differences. This looks like data overinterpretation?
13. Figure 5a and 6a are poorly described. No description is given in the legend as to what the box-plot markings mean? They should have non-parametric descriptors.
14. Figure 5a and 6a - individual measurements should be given, as quite standard for box-plots, were presented to help readers judge for themselves the data as it is not even clear what the N for each sub-event is, never mind the distribution of the individual data.
15. Figure 5a and 6a - whatever the error descriptors on plots represent, the Atlantic sub-event is highly variable –and is not different from Spain/France.
16. Figure 5a - the “Atlantic” is not statistically different from “Atlantic/Spain”, yet the argument in the text is that it is Spain that is the main originator of TMA, not the Atlantic?
17. Figure 5a – hard pressed to see how Spain/France differs from Atlantic/Spain based on the what can be garnered from data – yet the stats reports says they do?
18. Figure 6a – hard pressed to see how Atlantic differs from Africa/Med. based on the what can be garnered from data – yet

the stats reports says they do?

19. Figure 5b and 6b,c – error bars on graphs are stated to be standard deviations in legend – however - percentage data are not normally distributed and thus non-parametric descriptors should be used – which is difficult in this context. An appropriate test should be conducted on the duplicates, most reliably on concentration rather than percentage, and the outcome stated in the legend.

Version 1:

Reviewer comments:

Reviewer #1

(Remarks to the Author)

The reviewer thanks the authors for a rebuttal on a high level. The reviewer agrees with all answers and would recommend to publish this manuscript in this journal.

Reviewer #2

(Remarks to the Author)

The manuscript has been tidied-up by the authors in response to the reviewers wishes to improve its readability and interpretation. The rebuttal was thorough and many points were changed on the manuscript. However, I still have concerns that the dataset is relatively light in numbers of the key precipitation samples, in combination with air masses of clusters already being mixed in land and air contribution, and overlapping.

Here are my points:

1. Cluster 4, on which the interpretation of this paper focused, is labelled “Atlantic/Spain” (Figure 4), yet the authors interpret this as terrestrial. It seems, visually, that the Spain/France cluster 1 has a far greater terrestrial component to it than cluster 4, yet as a lower median TMAO precipitation content?
2. I note again that cluster 1 and cluster 4 have only an N of 4, from one sampling campaign, and I would be more convinced by their arguments based on low N if there were further campaigns that showed the same information.
3. Figure 5c points should have symbols to indicate which clusters they originate from – this would aid interpretation considerably, as the reader could then see if the cluster 4 points were the sole high ones, as from my visual interpretation at least one of these high points is from the Atlantic cluster 2, which somewhat confounds their thesis that DOC is of terrestrial origin, a point one which their thesis is strongly based – see line 258? It would also be highly useful for the reader if 95% CIs for the regression line were included for this graph as the correlation is relatively weak and influenced by a group of 4 points that fall outside another group of data with lower DOC's that has a lot of variance.
4. The correlations of arsenic species with specific groups of organic clusters for both TMAO origin seem important, line 268 onwards, yet we are only shown a summary of correlation P values, SI Table 4, with no presentation of the actual data - unless I am mistaken?
5. Perhaps the limited data presentation and interpretation is due to the number of parameters collected. It seems to me that their data is ripe for Principle Component Analysis interpretation, with Factor Analysis, and this would help clarify the interrelationship between the clusters, arsenic species and origin proxies such as DOC, organic speciation, elements and isotopes?
6. This brings me back my summary point 8 that while I think the manuscript is based on excellent analysis of hard to get samples, with a well written manuscript, and has the potential of show new insights, the hypothesis drawn from the data are based on both limited cluster N size, and limited data presentation (lack of symbol coding on scatter diagrams, reporting of P values with ought showing correlations). In their rebuttal the authors disagree with me on this point and state that “.. we stand confidently behind ... data analysis and its findings”. I disagree with them and think that more can be done to elucidate their data as outlined above.

Andy Meharg

Author response to referees' comments

We thank the two reviewers for their time reviewing our manuscript and their valuable comments, which significantly enhanced the clarity of the paper. We considered all comments carefully, and present our point-by-point responses below. The reviewers' comments are written in blue and the authors' responses are written in black. All page numbers given in the answers below refer to the revised manuscript in which changes are indicated using the track changes feature of word processor (i.e., revised manuscript with mark-ups).

Response to Reviewer #1:

This is a very interesting study which studies for the first time the origin of atmospheric methylated arsenic compounds and quantifies some deposition rates. This is really a missing piece in the puzzle. The foundation of the work is the excellent experimental design to analyse the air mass origin by using different markers in addition to sophisticated arsenic speciation methodology. I have not comment on the analytical methodologies. But before publication the authors should consider the following aspects to improve the manuscript.

We thank Reviewer #1 for this positive feedback on the novelty and quality of the work and for the constructive comments.

Especial aspects:

Line 77:

Yes the authors are correct that DMA(V) is taken up faster under certain circumstances. But does the exposure to toxic arsenic really increase? The legislation for arsenic in rice considers only inorganic arsenic and so far not methylated arsenic compounds. What is known about TMAO?? This statement should be revised.

We agree with the reviewer that legislation for rice currently only considers inorganic As (iAs) and so far not methylated As compounds. Nevertheless, MMAs^V and DMAs^V are classified as “possibly carcinogenic to humans on the basis of animal studies” (Group 2) by the International Agency for Research and Cancer (IARC)¹ and thus increase the exposure risk. IARC lists only one study for TMAs^{VO}, thus this species is classified with “limited evidence of carcinogenicity in experimental animals”¹.

We agree with the reviewer that the sentence was not clear enough and changed it to:

“For example, DMAs^V (possibly carcinogenic)¹ has a higher accumulation rate in rice plants due to more efficient transfer from roots to the grain than inorganic As species², increasing the potential dietary exposure risk.” (page 3, L81-84)

Line 147:

it is not clear how the errors are calculated. Standard deviations or std errors cannot be used, since 2 +/- 3 ng/L makes no sense... Please revise.

Thank you for pointing out that the standard deviations given in this paragraph were unclear. The reported standard deviations (e.g., 2 +/- 3 ng/L) referred to the variability in between events (and not to the measurement uncertainty) and were used to illustrate the high variability of As species concentrations in wet deposition samples. Measurement uncertainties were low, with an average relative standard deviation of duplicate measurements for individual As species in precipitation samples (in Fig. 5b and 6b,c) of 4±3%, 6±4% and 2±2% for TMAs^VO, DMAs^V and iAs, respectively.

We have now clarified the reported concentrations by indicating both concentration ranges and median values (instead of giving standard deviations that refer to the variability in-between events) as follows:

- For total and speciation data of collected precipitation samples, respectively:

“Total As concentrations in precipitation (sub)events ranged from 0.010 to 0.074 $\mu\text{g}\cdot\text{L}^{-1}$, resulting in total As deposition of 0.007-0.270 $\mu\text{g}\cdot\text{m}^{-2}$ (n=26, median: 0.038± $\mu\text{g}\cdot\text{m}^{-2}$; Supplementary Figure 1).” (page 4, L125-127)

“Concentrations in precipitation ranged from 1 to 3 $\text{ng}\cdot\text{L}^{-1}$ for DMAs^V (median of all sub-events: 1 $\text{ng}\cdot\text{L}^{-1}$), from 2 to 24 $\text{ng}\cdot\text{L}^{-1}$ for TMAs^VO (median: 4 $\text{ng}\cdot\text{L}^{-1}$), and from 6 to 52 $\text{ng}\cdot\text{L}^{-1}$ for iAs (median: 18 $\text{ng}\cdot\text{L}^{-1}$). These concentrations correspond to deposition ranges of 1-13 $\text{ng}\cdot\text{m}^{-2}$, 2-52 $\text{ng}\cdot\text{m}^{-2}$ and 4-194 $\text{ng}\cdot\text{m}^{-2}$ for DMAs^V and TMAs^VO and iAs (Fig. 1; Supplementary Table 2), respectively.” (page 5, L148-153)

- For total and speciation data of collected cloud water samples, respectively:

“In cloud water, total As ranged from 0.026-0.441 $\mu\text{g}\cdot\text{L}^{-1}$, corresponding to 0.001-0.009 $\text{ng}\cdot\text{m}^{-3}$ (n=56 sampled during 2019 campaign, median: 0.002 $\text{ng}\cdot\text{m}^{-3}$; Supplementary Figure 1)” (page 4, L127-129)

“Species concentration in cloud water ranged from 0.01 to 0.35 $\text{pg}\cdot\text{m}^{-3}$ for MMAs^V (median of all sub-events: 0.05 $\text{pg}\cdot\text{m}^{-3}$), from 0.02 to 0.50 $\text{pg}\cdot\text{m}^{-3}$ for DMAs^V (median: 0.16 $\text{pg}\cdot\text{m}^{-3}$), from 0.02 to 3.26 $\text{pg}\cdot\text{m}^{-3}$ for TMAs^VO (median: 0.95 $\text{pg}\cdot\text{m}^{-3}$) and from 0.1 to 7.7 $\text{pg}\cdot\text{m}^{-3}$ for iAs (median: 2 $\text{pg}\cdot\text{m}^{-3}$; Fig. 1), respectively.” (page 5, L156-162)

Line 154:

variable “as in the concentrations of inorganic and methylated As species” is that so, but more important is that the proportion of methylated and inorganic arsenic is variable and that is more important... I think there are not enough events measured that we can assess that terrestrial and marine have different concentrations... but proportions can be discussed....

We agree with the reviewer and adapted the sentence on page 5, L166-168:

“Overall, our results indicate a high variability in terrestrial and marine moisture sources as well as in the proportion of inorganic and methylated As species in wet deposition samples.”

Line 189:

the not quantitative recovery might be similar to previously published data, but that does not make it good. This should further been discussed, since that seems important for the entire calculations (short explanations should be given in the main text especially about the recalcitrant arsenic)

There seems to be a misunderstanding (see also the answer to the following comment). Indeed, the 1% HNO₃ extract did show quantitative recoveries, with an average of 87±33% of As species in aerosols being extracted (n= 70; total As refers to As measured in the concentrated HNO₃ + H₂O₂ digest). As described in lines 175-181 (pages 5-6), we used two types of extractions for As speciation analysis, i.e., water extract and 1% HNO₃ extract. The water extraction showed lower recoveries than the 1% HNO₃ (i.e., 38±17% of total As), which is expected given that water at circumneutral pH desorbs less As from minerals or dissolve mineral particles (e.g. (oxy)hydroxides) than in acidic solutions. With respect to recoveries within the water-extracts, the sum of detected As species added up to the total As amount found in the water extracts (95±10%).

Line 181:

“likely including inorganic As-mineral and/or organic colloidal complexes” this is gibberish... I think organically bound arsenic should be solubilized by the acid. I think that is not convincing. As in minerals is more likely. Here data would be good. A pooled sample might be subjected to XRD could give some information, at least if it is amorphous or not. Here more meet on the bone please.

We fully agree with the reviewer that organically bound arsenic would be solubilized by the 1% HNO₃ extraction. Indeed, the referred statement “*with an insoluble fraction (in water) likely including inorganic As-mineral and/or organic colloidal complexes (as commonly observed in soils and surface waters)*” refers to the water extract and not the 1% HNO₃ extract.

To avoid confusion, we changed the sentence as follows:

“Inorganic As was the dominant species in both extracts (water extract and 1% HNO₃), however, the water extract had lower proportions of inorganic As than the 1% HNO₃ extract, which is likely related to an insoluble recalcitrant As fraction (in water) potentially containing As-mineral and/or organic colloidal complexes (as commonly observed in soils³ and surface waters⁴).” (page 6, L193-198)

We also agree with the reviewer that the mineralogy of aerosols is a very interesting question that could be investigated in future studies. However, for these analysis, relatively large sample masses are necessary (in the mg-range^{5, 6}). For our collected aerosol filters, XRD analyses would be very challenging due to low aerosol masses at this remote location. To ensure that sufficient sample mass is obtained for such type of analyses, future field campaigns may target entire filters sampled over longer periods. However, this would jeopardize high-resolution sampling. We included a sentence providing such a perspective, as follows:

“Future studies are needed to better characterize the recalcitrant As fraction in aerosols. However, this would require the development of specific extraction methods and/or the use of solid phase speciation techniques, which require high element content and are therefore not easily applicable to aerosols collected in remote areas and/or with high temporal resolution.” (page 6, L198-202)

There is an elegant way to determine the origin of the trapped atmospheric samples, but it is not clear how the authors calculate the error used for the classification of the samples with regards to their origin.

Thank you for this question. The standard deviation of contributing moisture sources (page 7, L219-231) refers to the variability of moisture sources of individual sub-precipitation events within identified clusters. The respective standard deviations of defined source regions are given in the Supplementary Table 3. This was previously indicated in the manuscript as follows: “Averages and standard deviations of moisture sources for each cluster are given in Supplementary Table 3”. To make it clearer, we revised this sentence as follows: “Averages and standard deviations of moisture sources of individual sub-precipitation events within each cluster are given in Supplementary Table 3” (see page 7, L219-222).

Moreover, as described in Sodemann et al. (2008), the accuracy of trajectory calculations decreases with increasing calculation time. Trajectory calculations beyond 10 days can have high uncertainties due to wind field errors that lead to deviations from the actual movement of air parcels⁷. Improvements have been achieved thanks to higher resolution in time and space of reanalysis products⁸ and trajectory-calculation strategies⁹. In this study, we used an ensemble of 7-day backward trajectories with the recent ERA5 dataset from the European Centre for Medium Range Weather Forecasts. With this approach, we expect to be able to cover a large share of the sources given the climatological water vapour residence time estimates for the Pic du Midi region⁹.

Line 292:

The authors are correct to say that DMA(V) is the most important methylated species in the seawater. Here more discussion should be done, why this is? Algae metabolism should be described here...

We thank the reviewer for this suggestion, which we implemented in the revised manuscript as follows:

“Biomethylation of inorganic As has been proposed as a potential detoxification mechanism, in which As^V is taken up by marine algae, reduced to As^{III} and subsequently methylated to the less toxic species DMAs^V within the cell, which can then be excreted into the surrounding seawater¹⁰.” (page 10, L322-325)

We did not go into more detail on the algae metabolism beyond this sentence because of word limitation (and algae metabolism is out of the scope of this study).

Line 299:

Do we expect AsH₃ biogenic or anthropogenic generation ?? There are no volcanoes in Spain? Is this argument really relevant?

What we mean to say here is that inorganic As is likely the end product of different emitted As species (including AsH₃) from both anthropogenic and biogenic emission sources.

Indeed, AsH₃ has been linked to volcanic emissions^{11, 12}. In addition, AsH₃ has been associated with different biogenic and anthropogenic sources, including, emissions from peat soils^{13, 14}, biogas and sewage plants^{15, 16} as well as industrial emissions associated with the purification of liquefied petroleum gas¹⁷ (ref. 17 specifically includes one sampling location in Spain) or industrial fires¹⁸. Although these studies, particularly those for soils and biogas, did not

consistently report AsH₃ as the main emitted species, the contribution of AsH₃ to inorganic atmospheric As cannot be excluded.

For clarification, we adapted the text as follows:

“The proportions of inorganic As, which is the most abundant form of As, did not vary between clusters for precipitation (sub)events characterized by different moisture sources, except for cluster 4, which has relatively less inorganic As (Atlantic/Spain; Supplementary Figure 10). The small variations in different clusters could be related to the fact that inorganic As is the final oxidation product of various emitted volatile As species from both anthropogenic and biogenic processes (e.g., As^{III}₂O₃, As^{III}H₃).” (page 10, L327-334)

Line 347:

what is meant by anthropogenic methylated arsenic here?? Unclear do the authors mean herbicide usage or what??

In this sentence *“Anthropogenic emissions have decreased in the last decades, particularly in the Northern Hemisphere, but the identified biogenic methylated As species at Pic du Midi may still originate from anthropogenically derived As that has been re-emitted via biogenic processes.”*, we are not referring to anthropogenic methylated arsenic.

The sentence is referring to legacy As, which describes As that has originally been released into the environment by anthropogenic activities (such as coal burning, mining), but which has then undergone further transformation, e.g., inorganic As deposited from coal burning that is subsequently taken up by an organism and then biomethylated and volatilized. For clarification, we have added the following specification:

“Anthropogenic emissions have decreased in the last decades, particularly in the Northern Hemisphere, but the identified biogenic methylated As species at Pic du Midi may still originate from anthropogenically derived As from various sources (e.g., mining, coal burning, wood preservatives, herbicides), that have been re-emitted via biogenic processes.” (page 11, L379-383)

One aspect the reviewer misses. What is the stability of the individual methylated arsenic compounds in the atmosphere? What do we know and would that have a significant influence on the proportion of the individual methylated arsenic compounds? Do we know if demethylation can take place in the atmosphere in presence of OH radicals? Could that give a bias in the interpretations what the origin of the individual is...

Previous research on the stability of gaseous As species determined half-lives of approximately 8 h for methylarsines and longer for inorganic arsines under daytime conditions and up to weeks when tested in the dark¹⁹. We have added this information in the introduction with the following sentence: *“Volatile As has an estimated half-life of approximately 8 h (day-time conditions) for methylarsines and even longer for inorganic arsines and in the dark¹⁹.”* (page 2, L62-64)

Arsine and methylated arsines are expected to convert to non-volatile species following hydroxyl radical and/or O₂ addition based on computational studies of the gas phase As species' reactivity¹⁹. The formed pentavalent non-volatile species (i.e., MMAs^V, DMAs^V and TMAs^VO) are expected to undergo gas-particle transformation^{19, 20}. The influence of atmospheric processing on different As species have not been specifically investigated and

would require experiments in controlled smog chambers that involve both gas phase and condensed phase processes including aerosol and cloud chemistry.

Demethylation has not been studied in the atmosphere and, to the best of our knowledge, has only been reported under very specific environmental conditions, i.e., anoxic to methanogenic redox conditions in paddy soils^{21, 22}.

We agree that species stability and demethylation are very interesting aspects and should be studied in future work.

Response to Reviewer #2:

Summary

1. This is an impressive dataset of methylated arsenic species in troposphere precipitation, based on state-of-the-art analysis of difficult to obtain samples. It should add new insights to a field that is both emerging and of great consequence for arsenic biogeochemical cycling. A cycle of importance due to exposure to humans being of concern at trace concentrations of this element.

We thank Reviewer #2 for taking the time to read our article and acknowledging the broad and high-quality datasets we acquired.

2. However, the text over exerts it's priority in the field.

We assume that it refers to the points raised in the detailed comments 1 and 2, which we address in detail below.

3. While precipitation is event orientated the authors then go an pair this with aerosol data which is collected over a week. For modelling event air trajectory origins this pairing is not appropriate.

We did not pair the collected aerosol data with trajectory based moisture sources. The pairing of samples with moisture sources was solely done for wet deposition samples, i.e., precipitation and cloud water samples collected during the 2019 campaign (see page 25, Fig. 1). We specifically used this event-based sampling approach to enable the investigation of source proxies (chemical speciation, isotopes, elemental composition) within individual precipitation or cloud water events and paired them with moisture sources calculated specifically for these (sub)events.

In turn, the aerosol series (2015-2020) was used to obtain more long-term insights, however at lower temporal resolution (weekly sampling in contrast to event-based sampling of wet deposition samples).

To clarify the pairing with trajectory based moisture sources, we made the following changes in the manuscript:

- In the first paragraph of the subsection “Identification of distinct contributing moisture sources and source indicators” we modified the text as follows:

“To better understand the origin of methylated As species in atmospheric deposition, we investigated the relationship between As species and contributing moisture sources (in (sub) event-based collected wet deposition samples) as well as different chemical proxies of atmospheric sources to distinguish between marine vs. terrestrial and biogenic vs. anthropogenic sources (in both (sub) event-based collected wet deposition and weekly-aerosols samples).” (page 6, L208-212)

- We clarified the method section on air mass back trajectory and moisture source analysis: *“7-day backward trajectories at 9 verticals levels between 900 and 300 hPa from 5 points at and around Pic du Midi were calculated at 1-hour resolution for precipitation and cloud water samples and at 6-h resolution for the 2015-2020 aerosol time series. For the precipitation and cloud water samples, these trajectory datasets were used to identify evaporative moisture source regions (i.e., by tracking specific humidity changes along the trajectory) following the Lagrangian approach developed by Sodemann et al.⁷³. This moisture sources analysis was not performed for the aerosols series.”* (page 16, L539-550)

4. Origin groupings are confused and overlapping, and appear to have low N.

We assume that this comment refers to the points raised in detailed points 3, 9, and 12, which we address in detail below.

To briefly clarify this point here, the grouping of the 4 clusters is made as follows (as indicated in the discussion and method sections on page 7, L219-231 and page 16, L554-561, respectively)

- Moisture sources of individual sub-events are calculated. For example, the sub-event P17.1 has 77% moisture derived from Atlantic and 13% from Spain (see moisture source of precipitation and cloud water sub-events in Fig. 1 of the manuscript).
- Hierarchical agglomerative cluster analysis is applied to the individual moisture source dataset (i.e., moisture source data for each collected wet deposition sample) using Ward’s linkages based on Euclidean squared distances (see Figure A, panel a below).
- The first 4 identified clusters (i.e., the 4 groups of samples that show the most significant differences; see Figure A below) were used for further statistical analysis.

Figure A: Hierarchical agglomerative cluster analysis of individual moisture source dataset using Ward's linkages based on Euclidean squared distances. Panel (a) shows the hierarchical clustering dendrogram generated using the Ward linkage method. The dendrogram provides a visual representation of how the cases in the dataset (y-axis; here precipitation sub-events) are grouped into clusters based on their similarities in moisture sources. The distance of split or merge (i.e., cluster distance) is shown on the x-axis. Panel (b) shows the membership of individual sub-precipitation events for solutions of 2 to 6 clusters.

We agree with the reviewer that the four identified clusters of moisture sources have some overlap in source regions (Fig. 4a). Due to the atmosphere's dynamic nature, a process and/or source can be relevant for more than one cluster, but each cluster represents a unique combination of processes and sources. It should be noted that clusters were used to summarize general trends, and their statistical differences were determined using a suitable statistical tool (i.e., non-parametric Mann-Whitney-U test). However, in-depth calculations, including correlation coefficients, are derived for individual wet deposition samples, their chemical signatures (e.g., major element concentrations, S isotopes and speciation), and moisture source information for that specific (sub)event (please see answer the summary comment 5 and detailed comments 9 and 11). To emphasize this point, we have added the sample size to all reported Spearman rank correlations for the discussed speciation data of TMAs^VO on page 8, L256-261 and DMAs^V on page 9, L299-302 and L311-317. All statistical tests were reported with significance levels (p-values), which by definition considers the sample size.

5. The origin data portioning appears imprecise and over-interpreted.

We disagree with the reviewer that the moisture source apportionment is imprecise. The modelled moisture source data have an inherent uncertainty (see also the answer to the general comment 4 concerning the use of moisture sources and to the detailed comments 3, 9, 11 and 12). However, it is worth pointing out that our sampling design was deliberately chosen to keep sampling time and thus period of moisture source modelling as short as possible in order to identify more spatially restricted moisture source regions than for example if a precipitation sample would have been collected over multiple days. The range of sampling times of precipitation (sub-)events ranged between 10 and 900 min, and for cloud water (sub-)events it ranged between 25 and 160 min. Furthermore, the source signature of each

individual As species in each individual sample were investigated using a combination of various chemical measurements and modelling. We did not only use moisture source diagnostics but also additional (independent) chemical data sets including S isotopes, S speciation, elemental concentrations (e.g., Na contents), dissolved organic carbon, and organic aerosol composition to study source characteristics.

We want to emphasize again that the pairing of modelled and chemical data was done for wet deposition samples of the high-resolution campaign. We specifically used this (sub)event-based sampling approach to enable a more detailed source characterization for wet deposition samples. In other words: the shorter the time scale of sampling - the more detailed the source apportioning. Moreover, similar source signatures were identified in both investigated datasets of atmospheric deposition samples (2-month campaign of precipitation and cloud water samples and a 5-year aerosol series), which further supports the validity of the results on both short and long-term scales.

Finally, as due to the inherent dynamic nature of the atmosphere, to some extent overlap of different processes and sources will occur. Therefore, in our study we solely discuss *dominant* source signatures of As species.

6. Statistics throughout are muddled – see below.

This general statement seems to refer to the detailed comments 13-18, which are about the box-and-whisker plots and which we address in detail below.

In this study, we used commonly used and robust statistical approaches, i.e., hierarchical agglomerative cluster analysis (for the moisture source dataset) using Ward's linkages based on Euclidean squared distances, bivariate correlations performed as Spearman rank correlations (with determination of significance levels (2-tailed)), and significant difference analysis using Mann-Whitney U-test (with determination of significance levels). Both the Spearman rank correlation and the Mann-Whitney U-test are non-parametric and thus do not require a normal distribution of the data.

These various statistical analyses were described in the original manuscript in different subsections of the results and discussion section and/or of the method section. To improve clarity, we have now described all statistical analyses together in the method section as follows:

“Statistical analyses. All statistical analyses were performed using IBM SPSS Statistics 26. Correlations were performed as Spearman rank correlations (correlation coefficient indicated by r_s), and significance levels (2-tailed) are indicated by $p < 0.05$ or $p < 0.01$. For the collected precipitation events ($n=26$), we performed hierarchical agglomerative cluster analysis on the moisture source datasets using Ward's linkages based on Euclidean squared distances to identify their main source regions in terms of dominant contributing moisture sources. Then, for the main variables of interest in this study (e.g., proportions of As species), significant differences between the identified clusters were investigated using the Mann-Whitney U-test with significance levels (2-tailed) indicated by $p < 0.05$ or $p < 0.01$.” (page 16, L551-561)

7. Given the above, a key thesis of this report that trimethyl arsine is derived primarily from terrestrial environs cannot be supported.

This point was also raised in the detailed comment 10, which we address below. We agree with the reviewer that the last paragraph of the TMAs^{VO} section was not clear enough. Indeed,

we did not mean to imply that TMAs^{VO} is solely of terrestrial origin, and therefore rephrased the last paragraph of the TMAs^{VO} section, as follows:

“Overall, we found strong links between TMAs^{VO} and terrestrial source proxies, but no links with marine proxies. These results indicate that terrestrial biogenic sources are more important than marine sources at Pic du Midi. Nevertheless, this does not mean that potential marine sources of TMAs^{VO}, as previously suggested^{23,24}, can be excluded. Marine sources are likely not recognized due the higher dominance of terrestrial sources at our sampling site.” (page 9, L289-293)

Our study does not intend to invalidate or question previous work that suggested mainly marine sources of TMAs^{VO} at a different study site. Marine sources are certainly likely but probably not recognized in our dataset due to relatively higher importance of terrestrial sources for TMAs^{VO} at our sampling site.

8. The data analysis needs revisiting, and to be presented in a coherent and robust form so that a new thesis may emerge from that analysis.

We disagree with this point. All steps in our study, from sampling, analytical workflows, data analyses, and integration of modelled and measured data were carefully planned and thoroughly executed.

In summary:

- We collected different atmospheric deposition samples, including precipitation (n=26), cloud water (n=56) and aerosols (n=134), at a location which is influenced by long-range elemental transport from both continental (Europe, Africa) and marine (Atlantic Ocean and Mediterranean Sea) regions²⁵, in addition to transport from local sources.
- We used different chemical measurements and state-of-the-art atmospheric transport modelling approaches, integrating chemistry and atmospheric dynamics to gain insights into the sources of As species in atmospheric deposition.
- In addition to As speciation obtained by improved pre-concentration and speciation analysis by LC-ICP-MS/MS, various chemical analyses were performed included total concentrations of (trace) elements, S speciation, S isotopes, dissolved organic carbon, and organic aerosol composition. These analyses were performed with quality control and replicate measurements (when possible).
- These chemical analyses were carefully paired with dominant moisture sources and atmospheric transport patterns (considering the exact sampling time and period) that were estimated using three-dimensional wind fields from the atmospheric reanalysis dataset ERA5.

This approach allowed for the identification of source signatures of methylated As species in atmospheric deposition based on the statistical analysis between As speciation and other independent dataset of chemical source proxies and atmospheric transport patterns. Performed statistical analyses were done correctly and using the appropriate methods (see general comment 6).

Therefore, we stand confidently behind all the steps presented above, including data analysis and its findings.

In case the reviewer with this comment specifically addresses our conclusion of a dominant terrestrial source of TMAs^{VO}, as stated in the answer to the general comment 7, we rephrased this conclusion. Although based on our datasets only a terrestrial source signature could be identified, we agree with the reviewer that marine contributions of TMAs^{VO} are possible.

Detailed points

1. Regarding priority, line 23

of the abstract states - "However, the speciation of methylated arsenic in deposition and their potential long-range transport is unknown." Reference 18 in their citation list, was the first to identify methylated arsenic species in precipitation. The long range transport of methylated arsenic species was first explored by reference 17 in the citation list. Other studies have modelled the back trajectories for methylated arsenic species in precipitation, refs 19 and 20 on their reference list. Perhaps the authors mean that they are first to measure methylated species and their long-distant transport in tropospheric samples?

We agree with the reviewer that previous studies have measured methylated As species in atmospheric deposition samples as stated in the introduction in L67-68: "*While several studies have investigated the occurrence of methylated As species in atmospheric deposition*^{17, 18, 19, 20, 21, 22, 23}, *the degree to which long range transport of variable sources can contribute to their deposition is not well described*".

For clarification, we changed the referred sentence as follows:

"However, the range of transport and source signature of arsenic species remain understudied." (page 1, L23-24)

2. The line 23

statement regarding priority is contradicted by line 66 - "While several studies have investigated the occurrence of methylated As species in atmospheric deposition^{17, 18, 19, 20, 21, 22, 23},"?

See previous answer on detailed comment 1.

3. Line 67

"the degree to which long- range transport of variable sources can contribute to their deposition is not well described." This is a confusing statement. Does it mean that the scientific protocols described in previous studies were not clear, or perhaps more likely, that they think their approach is more sophisticated? Please clarify, noting while an incremental improvement in methodology presented here does not invalidate historic studies on which their manuscript builds.

We very much value previous studies on atmospheric As and accordingly referenced them both in the introduction and discussion sections. With the sentence "*While several studies have investigated the occurrence of methylated As species in atmospheric deposition*^{20, 24, 26, 27, 28, 29, 30}, *the degree to which long-range transport of variable sources can contribute to methylated As deposition is not well-described*", we did not want to imply that previous studies were unclear nor did we want to invalidate them. Previous studies on As speciation in atmospheric deposition were performed in (sub-)urban/rural^{20, 24, 27, 28, 29, 30} and forest areas²⁶ which offer very valuable insights into specific emission sources at a local to region scale, however are not suitable to investigate the range of transport and contribution of source signatures of individual As species (see also reply to following comment 4).

Concerning the sampling strategy and trajectory based techniques, our approach indeed represents an improvement to previous (including our own) work that combined trace element analyses with modelled air mass trajectories. In our previous study by Blazina et al. (2017), we applied 72 h backward trajectories to investigate the sources of different trace elements³¹.

Backward trajectories give insights into potential source regions however, the selected time period of analysis (in that case 72 h) co-dictates which source regions are identified. Furthermore, for precipitation samples, it is important to identify the source regions of atmospheric moisture, in addition to air masses, which cannot be done with air mass trajectories alone. The use of moisture source diagnostics, which is an established method in atmospheric water cycle research^{7, 32} that has been applied in a wide range of studies published in the last decade^{33, 34, 35, 36, 37, 38, 39, 40}, enables the identification of evaporation sites and thus direct interaction of moisture with the boundary layer along the trajectory. In addition, each uptake location is weighted according to its contribution to the final humidity of a trajectory that leads to the formation of the sampled precipitation event. If a decrease in specific humidity (i.e., precipitation) occurs along a trajectory after one or more uptakes, the contributions of previous uptakes are discounted. For a given time step, the moisture source contribution of the different trajectories is weighted according to their final specific humidity loss (i.e. their contribution to precipitation at the sampling location). Therefore, when it previously rained along the atmospheric air parcel trajectory, this is not included anymore as a moisture source.

In summary, moisture source diagnostics represents a more precise approach compared to the previous approach applied in Blazina et al. (2017) and other papers focused on methylated As species that used a similar approach, as moisture uptakes and losses are taken into account, thus enabling a more realistic estimation of specific distances of investigated air parcels.

For clarification, we revised the referred sentence as follows:

“While few studies have investigated the occurrence of methylated As species in atmospheric deposition and their general source regions^{20, 24, 26, 27, 28, 29, 30}, the degree to which long-range transport of variable sources can contribute to their deposition is still unclear.” (page 2-3, L69-71)

4. Line 69:

“Previous studies ... low altitude sites ...” Note also that low altitude and urban sites are perhaps more relevant to actual deposition of arsenic than what is in the troposphere, and thus complimentary to tropospheric studies. It would be good to contrast and compare positives and negatives of both approaches.

We agree with the reviewer that low altitude and urban sites are relevant when investigating As deposition. For example, anthropogenic emissions of local point sources can deposit substantial amounts of As to nearby agricultural systems (e.g., ^{41, 42}). However, the specific objective of this study was to investigate long-range atmospheric transport of As at a remote site away from major emissions sources, which makes low-altitude and urban sites not suitable for our goal. As stated in the second part of the referred sentence, one major challenge of studies conducted at low altitude sites is that they are located within the atmospheric boundary layer and therefore mainly influenced by local processes, which can obscure source signatures that travelled over longer distances. Nevertheless, by no means do we intend to invalidate or question results from studies that had a different research focus than ours. Therefore, we feel that it is not required in our study to compare “positives and negatives” as different studies have different objectives, which we now clarified in the referred sentence:

“Previous studies aimed to investigate local to regional sources by sampling in (sub-)urban/rural^{20, 24, 27, 28, 29, 30} or forest areas²⁶. To specifically study long-range transport, locations that are frequently exposed to free tropospheric air with limited influence of local processes (such as at high-altitude) are most suitable.” (page 3, L72-77)

5. Line 90:

The authors state that aerosol samples are collected weekly – this seems quite out of sync with the resolution of the precipitation data, stated to be “high temporal resolution”, line 85, with which the aerosol data is paired for modelling?

Please, see our reply to summary comment 1 above.

6. Line 136:

states: “This analytical advancement” Their lyophilization is both obvious and relatively standard. The “advanced” analysis is the arsenic speciation which was developed by others before them. Thus, while their methodology is an incremental, it is useful, improvement with respect to lowering limits of detection.

We agree with the reviewer that lyophilisation is a common pre-concentration technique. However, to the best of our knowledge, this technique has not been used for the analysis of As species in atmospheric deposition samples. Accordingly, the extraction of As species from aerosol samples and their pre-concentration via lyophilisation, while preserving the integrity of the native speciation, required careful systematic testing (detailed description in Supplementary Discussion 1). In addition, the combination of pre-concentration and optimization of the LC-ICP-MS/MS method (detailed description in Supplementary Discussion 1) improved detection limits, which was necessary to analyse the collected atmospheric deposition samples in this remote setting.

We agree with the reviewer that “analytical advancement” was overselling and thus adapted the referred sentence:

“This improvement in detection limits allowed the quantification of TMA_s^V and DMA_s^V in all collected precipitation samples in contrast to most previous studies (DMA_s^V being below or equal to detection limits in >99%^{24, 26, 27} and TMA_s^V in >67% of the samples analysed in previous studies^{24, 26}).” (page 5, L142-146)

7. Line 136

states - “allowed the detection of methylated As species in all collected samples in contrast to most previous studies.”. Does this mean that all samples should all samples contain (detectable) methylated arsenic species? They do not detect all methylated species in all samples – see line 142 where MMA was only detected in two samples. What I think the authors are trying to say is that they have improved limits of detection in their arsenic speciation analysis, which is correct, but they have already stated this?

To enable the investigation of source information of individual methylated As species, particularly using statistical analyses, sufficient detection limits are absolutely necessary, otherwise a substantial bias is introduced. Since MMA_s^V was only detected in two precipitation samples, this species was excluded from statistical analysis, as stated on page 10, L338-339: *“ MMA_s^V was only detected in two precipitation events and therefore not included in the cluster analysis.”*

To us, it was important to not only improve the method’s detection limits but also to evaluate whether these improved detection limits offered new insights. For this purpose and in light of the following detailed comment 8, we have now refined the comparison of detected methylated As species with previous studies by calculating the proportions of precipitation samples in which methylated As species were detected in previous studies (in % of total analysed samples in respective studies). The output of this comparison is now included in the revised sentence:

“This improvement in detection limits allowed the quantification of TMAs^VO and DMAs^V in all collected precipitation samples in contrast to most previous studies (DMAs^V being below or equal to detection limits in >99%^{24, 26, 27} and TMAs^VO in >67% of the samples analysed in previous studies^{24, 26}).” (page 5, L142-146)

8. Line 136

states - “ most previous studies.” Here “most” is vague. It would be good to know which studies reached the standard of theirs?

Please, see the answers to detailed comments 6 and 7 above as the referred sentence has been revised, and includes now a more careful comparison with the literature.

9. Line 201

states that 26 events had their sources identified, with only 4 of these from Spain. “N” here is important as the key conclusion of this report is that samples from this region are elevated in percentage TMAsO compared to marine regions?.

The identified dominant terrestrial source signature of TMAs^VO in precipitation collected at Pic du Midi was not only based on moisture source uptakes over Spain (moisture source plots in Fig. 4a), but also on a range of other (independent) parameters, including dissolved organic carbon, S isotopes, S speciation, and Na content in. In addition, the dominant terrestrial source signature of TMAs^VO was further supported with significant correlations between TMAs^VO proportions and specific terrestrial organic compounds (including pyrolytic products of carbohydrates) in the long-term aerosols series, which further supports the validity of the results on both short and long-term scales.

We agree that cluster 4 of the moisture source data set has a small sample size. However, the discussed correlations to chemical source proxies were calculated for *all* collected precipitation samples (n=26), not solely within each cluster (see also described in our reply to summary comment 4). To emphasize this point, we have added the sample size to all reported Spearman rank correlations for discussed speciation data of TMAs^VO (page 8, L256-261 and caption of Fig. 5b) as well as all other Spearman rank correlations presented in the manuscript (page 9, L299-302 and L311-317; and caption of Fig. 6b, c).

10. Line 226

- paragraph starting at - “TMAsO has a dominant terrestrial biogenic origin” does not discuss that the only study reporting arsines generation from seawater, reference 13 on their list, found that TMA was, by far, the dominant species produced. This seems somewhat pertinent do the discussion as it is in opposition to their manuscripts primary finding?

The referred study by Savage et al. (ref. 13 in manuscript) tested a seawater sample collected from Belfast Lough (Irish Sea), which was incubated in 8 replicates. Indeed, trimethylarsine was reported as the primary biovolatilized product, with dimethylarsine also being observed when DMAs^V was spiked into tested seawater²³.

Other studies that analysed As speciation in natural seawater, found DMAs^V as the most abundant methylated As species⁴³, as described in L318-319. This is further supported by the two studies by Wurl et al (2013, 2015), that reported DMAs^V as the most abundant methylated As species in surface waters of two large transects collected in the North Atlantic^{44, 45}.

Based on previous studies, it is thus plausible that both DMAs^V and TMAs^{VO} can be present in marine environments. The relative abundance of these species will likely depend on many factors including the microbial community and environmental conditions⁴⁶ and may thus vary both temporarily and spatially in the Atlantic and other marine systems.

As described in our reply to summary comment 7 and later to detailed comment 16, our data does not imply that marine sources do not have an influence on TMAs^{VO}; it could merely indicate that terrestrial sources are more dominant at our study site.

For clarification, we made the following changes to the TMAs^{VO} section:

- In the first paragraph, that discusses the variability of TMAs^{VO} proportions in between clusters, we modified the text as follows:
“Still, it should be noted that although cluster 2 (Atlantic) has significantly lower TMAs^{VO} proportions than cluster 4, TMAs^{VO} is still the main methylated As species in this cluster and therefore, Atlantic background sources of TMAs^{VO} are likely.” (page 8, L252-255)
- Furthermore, the concluding paragraph was changed as follows:
“Overall, we found strong links between terrestrial source proxies and TMAs^{VO} and no links with marine proxies. These results indicate that for Pic du Midi terrestrial biogenic sources are more important than marine sources. Nevertheless, this does not mean that potential marine sources of TMAs^{VO}, as previously suggested^{23, 24}, can be excluded. Marine sources are likely not recognized at our sampling site due the higher dominance of terrestrial sources.” (page 9, L289-294)

11. Line 230

Spain and Portugal and noted as a source region, but Portugal then disappears from the text? The text seems to indicate that the mountainous regions of N. Spain and S. France are important, not the whole of Spain – but the text and presentation is confused.

See also answer to general comment 4 and specific comment 9.

The discussed correlations between TMAs^{VO} proportions and moisture uptakes over Portugal were calculated for all collected precipitation samples (not solely within each cluster). To emphasize this point, we have added the sample size to all reported Spearman rank correlations for discussed speciation data of TMAs^{VO} as follows:

“In particular, sources originating from Portugal and Spain are significant according to positive correlations between moisture sources of these regions and TMAs^{VO} proportions (i.e., respectively, $r_s=0.399$ and $r_s=0.408$, $n=26$, $p<0.05$).” (page 8, L256-258)

Moreover, it should be noted that the clusters shown in Fig. 4 have an abbreviated name according to their two dominant moisture source regions. The highest moisture uptakes over Portugal were observed in cluster 4 (as shown in detail in Supplementary Table 3).

12. Figure 4a

- the clusters all overlap, all with their epicentre in N.W. Spain, yet much is made of these cluster differences. This looks like data overinterpretation?

Moisture uptakes in identified clusters are displayed as % uptakes per area (shown in colour scale 0%–28% per 10^5km^2) in respective maps (Fig. 4a), which means that areas indicated in red have high moisture uptakes per area but if the area is small, they are not necessarily predominant sources. For example: in cluster 2 (Fig. 4a), the N. Atlantic has a relatively small moisture uptake per area (so it is indicated in blue), but because the area is very large, it still has dominant contribution to the overall moisture uptake.

The maps in Fig. 4a solely serve as illustration of calculated moisture source regions. For any statistical analysis (hierarchical cluster analysis and correlations) done, specific contributions of source regions were not taken from the maps but from calculated percentages for each defined moisture source area (see Supplementary Table 3). As we do not interpret the data based on the maps there is no data over interpretation as suggested by the reviewer.

To clarify this point, we added the following specification in the caption of Fig. 4:

“Moisture uptakes are displayed as percentage uptakes per area (shown in colour scale 0%–28% per 10^5 km^2), thus depending on the defined size of areas, high moisture uptakes (red) do not necessarily correspond to predominant sources. Specific contributions of moisture uptakes in pre-defined regions within identified clusters are listed in Supplementary Table 3.”

13. Figure 5a and 6a

are poorly described. No description is given in the legend as to what the box-plot markings mean? They should have non-parametric descriptors.

We agree with the reviewer that a detailed description of the box plot was missing and thus added the specification of the box-plot in the captions of Fig. 5 and Fig. 6, as follows:

“The boxplots show the interquartile range, representing the middle 50% of the data, which fall between the upper quartile (75% data below that score) and the lower quartile (less than 25% below that score). The whiskers refer to the 5th/95th percentiles.”

14. Figure 5a and 6a

- individual measurements should be given, as quite standard for box-plots, were presented to help readers judge for themselves the data as it is not even clear what the N for each sub-event is, never mind the distribution of the individual data.

We agree with the reviewer. We have thus added the individual data points to all box plots presented in the manuscript (Fig. 5a and 6a). It should be noted that the concentration of As species with corresponding moisture sources of each (sub)-event (so each individual measurement) are presented in Fig. 1.

15. Figure 5a and 6a

- whatever the error descriptors on plots represent, the Atlantic sub-event is highly variable – and is not different from Spain/France.

Figures 5a and 6a are not bar charts with error bars but box plots with whiskers. Regarding the differences between clusters, we used the Mann-Whitney-U test to determine whether the proportions of each As species between clusters (shown in the box plots in Figure 5a and 6a)

are statistically different or not. This is represented with the letters “x” and “y” above the box plots in Fig. 5a and 6a, and was explained in the captions as follows “*Different letters (x, y) in (a) denote significance levels based on Mann-Whitney-U test, $p < 0.05$.*”.

The output of the Mann-Whitney-U test should be interpreted as follows:

- if the letters for 2 clusters are the same, the median of these 2 clusters are not statistically different (e.g. cluster 1:xy and cluster 2:x are statistically not different as both are denoted with an “x”)
- if the letters are different (e.g. cluster 2:x and cluster 4:y) the median of these 2 clusters are statistically different.

For example: for the Atlantic cluster 2 and the Atlantic/Spain cluster 4, TMAs^{VO} proportions are significantly different between these clusters as they are indicated by a different letter (x for cluster 2, y for cluster 4), and the associated p value is 0.02 (Fig. 5a)

For further clarification, we added the following sentence in the caption of Fig. 5a and 6a:

“For all clusters with the same letter, the difference between species proportion is not statistically significant”

16. Figure 5a

- the “Atlantic” is not statistically different from “Atlantic/Spain”, yet the argument in the text is that it is Spain that is the main originator of TMA, not the Atlantic?

We wrote that cluster 4 has significantly higher TMAs^{VO} proportions than cluster 2 and that this may indicate an important terrestrial source. We did not write that Atlantic moisture has no influence.

We then argued that TMAs^{VO} has a terrestrial source based on chemical proxy data, including i) significant link to dissolved organic carbon (for TMAs^{VO} in precipitation samples collected for 2 months on a sub-event basis); and ii) specific terrestrial organic compounds, including pyrolytic products of carbohydrates (for TMAs^{VO} weekly aerosol collected over 5 years). A dominant terrestrial source signature is further supported by no significant links to investigated marine proxies (i.e., S isotopes, S speciation, Na content in precipitation, and algae markers - e.g. proteins in aerosols).

17. Figure 5a

– hard pressed to see how Spain/France differs from Atlantic/Spain based on the what can be garnered from data – yet the stats reports says they do?

The two clusters Spain/France and Atlantic/Spain are not statistically different as they are both denoted with a “y” (please see our reply to detailed comment 15 on Figure 5a and 6a). Therefore, the “stats report” does not actually say that the Spain/France and Atlantic/Spain clusters differ in Figure 5a as suggested by the reviewer. We also did not write that they differ in the discussion section.

18. Figure 6a

– hard pressed to see how Atlantic differs from Africa/Med. based on the what can be garnered from data – yet the stats reports says they do?

The two clusters Atlantic (denoted with a “x”) and Africa/Med.Sea (denoted with a “y”) are statistically different based on the Mann-Whitney-U test. Please see also our reply to detailed comment 15 on Figure 5a and 6a.

19. Figure 5b and 6b,c

– error bars on graphs are stated to be standard deviations in legend – however - percentage data are not normally distributed and thus non-parametric descriptors should be used – which is difficult in this context. An appropriate test should be conducted on the duplicates, most reliably on concentration rather than percentage, and the outcome stated in the legend

The error bars indeed indicate standard deviations from duplicate measurements (i.e., measurement uncertainty) of individual deposition samples, expressed as percentage of the measured concentration. This is the standard way of presenting concentration data. We agree that the actual standard deviation (in units of concentration) should be given as well. Concentration values expressed in $\text{ng}\cdot\text{L}^{-1}$ and deposition values expressed in $\text{ng}\cdot\text{m}^{-2}$, each with corresponding standard deviations for each sample are now indicated in Supplementary Table 2.

The average relative standard deviation (also referred to as coefficient of variation) of duplicate measurements for individual As species in precipitation (shown in Fig. 5b and 6b,c) accounted for $4\pm 3\%$, $6\pm 4\%$ and $2\pm 2\%$ for TMAAs^{VO} , DMAAs^{V} and iAs , respectively, with an overall average of below 4%. It should be noted that shown standard deviations include the propagation of uncertainty associated to As species concentrations obtained in duplicates. The variability of uncertainty is primarily driven by samples with low concentrations, which are generally expected to have higher uncertainties.

For clarification, we added the following specification in the figure description of Figs. 5-6:

“the error bars represent the uncertainty of quantification by LC-ICP-MS/MS in duplicate (relative standard deviation of TMAAs^{VO} measurements: $4\pm 3\%$), invisible error bars indicate standard deviations within the symbol.”

References

1. IARC. Some drinking-water disinfectants and contaminants, including arsenic. *IARC Monogr Eval Carcinog Risks Hum* **84**, 1-477 (2004).
2. Zhao F-J, Zhu Y-G, Meharg AA. Methylated Arsenic Species in Rice: Geographical Variation, Origin, and Uptake Mechanisms. *Environmental Science & Technology* **47**, 3957-3966 (2013).
3. Tolu J, *et al.* Understanding soil selenium accumulation and bioavailability through size resolved and elemental characterization of soil extracts. *Nature Communications* **13**, 6974 (2022).
4. Neubauer E, v.d. Kammer F, Hofmann T. Using FLOWFFF and HPSEC to determine trace metal–colloid associations in wetland runoff. *Water Research* **47**, 2757-2769 (2013).
5. Queralt I, Sanfeliu T, Gomez E, Alvarez C. X-ray diffraction analysis of atmospheric dust using low-background supports. *Journal of Aerosol Science* **32**, 453-459 (2001).
6. Nowak S, Lafon S, Caquineau S, Journet E, Laurent B. Quantitative study of the mineralogical composition of mineral dust aerosols by X-ray diffraction. *Talanta* **186**, 133-139 (2018).
7. Sodemann H, Schwierz C, Wernli H. Interannual variability of Greenland winter precipitation sources: Lagrangian moisture diagnostic and North Atlantic Oscillation influence. *Journal of Geophysical Research: Atmospheres* **113**, (2008).
8. Hoffmann L, *et al.* From ERA-Interim to ERA5: the considerable impact of ECMWF's next-generation reanalysis on Lagrangian transport simulations. *Atmos Chem Phys* **19**, 3097-3124 (2019).
9. Sprenger M, Wernli H. The LAGRANTO Lagrangian analysis tool – version 2.0. *Geosci Model Dev* **8**, 2569-2586 (2015).
10. Zhang S-Y, Sun G-X, Yin X-X, Rensing C, Zhu Y-G. Biomethylation and volatilization of arsenic by the marine microalgae *Ostreococcus tauri*. *Chemosphere* **93**, 47-53 (2013).
11. Arndt J, Ilgen G, Planer-Friedrich B. Evaluation of techniques for sampling volatile arsenic on volcanoes. *Journal of Volcanology and Geothermal Research* **331**, 16-25 (2017).
12. Arndt J, Planer-Friedrich B. Moss bag monitoring as screening technique to estimate the relevance of methylated arsine emission. *Science of The Total Environment* **610-611**, 1590-1594 (2018).
13. Mestrot A, Feldmann J, Krupp EM, Hossain MS, Roman-Ross G, Meharg AA. Field Fluxes and Speciation of Arsines Emanating from Soils. *Environmental Science & Technology* **45**, 1798-1804 (2011).
14. Mestrot A, *et al.* Quantitative and qualitative trapping of arsines deployed to assess loss of volatile arsenic from paddy soil. *Environ Sci Technol* **43**, 8270-8275 (2009).
15. Weithmann N, *et al.* Arsenic metabolism in technical biogas plants: possible consequences for resident microbiota and downstream units. *AMB Express* **9**, 190 (2019).
16. Mestrot A, Xie W-Y, Xue X, Zhu Y-G. Arsenic volatilization in model anaerobic biogas digesters. *Applied Geochemistry* **33**, 294-297 (2013).

17. Hernández-Fernández J. Quantification of arsine and phosphine in industrial atmospheric emissions in Spain and Colombia. Implementation of modified zeolites to reduce the environmental impact of emissions. *Atmospheric Pollution Research* **12**, 167-176 (2021).
18. Griffiths SD, Entwistle JA, Kelly FJ, Deary ME. Characterising the ground level concentrations of harmful organic and inorganic substances released during major industrial fires, and implications for human health. *Environment International* **162**, 107152 (2022).
19. Mestrot A, Merle JK, Broglia A, Feldmann J, Krupp EM. Atmospheric Stability of Arsine and Methylarsines. *Environmental Science & Technology* **45**, 4010-4015 (2011).
20. Jakob R, *et al.* Atmospheric stability of arsines and the determination of their oxidative products in atmospheric aerosols (PM10): evidence of the widespread phenomena of biovolatilization of arsenic. *Journal of Environmental Monitoring* **12**, 409-416 (2010).
21. Chen C, *et al.* Sulfate-reducing bacteria and methanogens are involved in arsenic methylation and demethylation in paddy soils. *The ISME Journal* **13**, 2523-2535 (2019).
22. Chen C, Shen Y, Li Y, Zhang W, Zhao F-J. Demethylation of the Antibiotic Methylarsenite is Coupled to Denitrification in Anoxic Paddy Soil. *Environmental Science & Technology* **55**, 15484-15494 (2021).
23. Savage L, Carey M, Williams PN, Meharg AA. Biovolatilization of Arsenic as Arsines from Seawater. *Environmental Science & Technology* **52**, 3968-3974 (2018).
24. Savage L, Carey M, Williams PN, Meharg AA. Maritime Deposition of Organic and Inorganic Arsenic. *Environmental Science & Technology* **53**, 7288-7295 (2019).
25. Fu X, Maruszczak N, Wang X, Gheusi F, Sonke JE. Isotopic Composition of Gaseous Elemental Mercury in the Free Troposphere of the Pic du Midi Observatory, France. *Environmental Science & Technology* **50**, 5641-5650 (2016).
26. Huang J-H, Matzner E. Biogeochemistry of Organic and Inorganic Arsenic Species in a Forested Catchment in Germany. *Environmental Science & Technology* **41**, 1564-1569 (2007).
27. Savage L, *et al.* Elevated Trimethylarsine Oxide and Inorganic Arsenic in Northern Hemisphere Summer Monsoonal Wet Deposition. *Environmental Science & Technology* **51**, 12210-12218 (2017).
28. Tziaras T, Pergantis SA, Stephanou EG. Investigating the Occurrence and Environmental Significance of Methylated Arsenic Species in Atmospheric Particles by Overcoming Analytical Method Limitations. *Environmental Science & Technology* **49**, 11640-11648 (2015).
29. Tanda S, Gingl K, Ličbinský R, Hegrová J, Goessler W. Occurrence, Seasonal Variation, and Size Resolved Distribution of Arsenic Species in Atmospheric Particulate Matter in an Urban Area in Southeastern Austria. *Environmental Science & Technology* **54**, 5532-5539 (2020).
30. Tanda S, Ličbinský R, Hegrová J, Faimon J, Goessler W. Arsenic speciation in aerosols of a respiratory therapeutic cave: A first approach to study arsenicals in ultrafine particles. *Science of The Total Environment* **651**, 1839-1848 (2019).
31. Blazina T, *et al.* Marine Primary Productivity as a Potential Indirect Source of Selenium and Other Trace Elements in Atmospheric Deposition. *Environmental Science & Technology* **51**, 108-118 (2017).

32. Sodemann H. Beyond Turnover Time: Constraining the Lifetime Distribution of Water Vapor from Simple and Complex Approaches. *Journal of the Atmospheric Sciences* **77**, 413-433 (2020).
33. Winschall A, Pfahl S, Sodemann H, Wernli H. Comparison of Eulerian and Lagrangian moisture source diagnostics & the flood event in eastern Europe in May 2010. *Atmos Chem Phys* **14**, 6605-6619 (2014).
34. Aemisegger F, Pfahl S, Sodemann H, Lehner I, Seneviratne SI, Wernli H. Deuterium excess as a proxy for continental moisture recycling and plant transpiration. *Atmos Chem Phys* **14**, 4029-4054 (2014).
35. Suess E, Aemisegger F, Sonke JE, Sprenger M, Wernli H, Winkel LHE. Marine versus Continental Sources of Iodine and Selenium in Rainfall at Two European High-Altitude Locations. *Environmental Science & Technology* **53**, 1905-1917 (2019).
36. Thurnherr I, *et al.* Meridional and vertical variations of the water vapour isotopic composition in the marine boundary layer over the Atlantic and Southern Ocean. *Atmos Chem Phys* **20**, 5811-5835 (2020).
37. Aemisegger F, *et al.* How Rossby wave breaking modulates the water cycle in the North Atlantic trade wind region. *Weather Clim Dynam* **2**, 281-309 (2021).
38. Aemisegger F, *et al.* Fingerprints of Frontal Passages and Post-Depositional Effects in the Stable Water Isotope Signal of Seasonal Alpine Snow. *Journal of Geophysical Research: Atmospheres* **127**, e2022JD037469 (2022).
39. Villiger L, Aemisegger F. Water isotopic characterisation of the cloud–circulation coupling in the North Atlantic trades – Part 2: The imprint of the atmospheric circulation at different scales. *Atmos Chem Phys* **24**, 957-976 (2024).
40. Breuninger ES, *et al.* Influences of sources and weather dynamics on atmospheric deposition of Se species and other trace elements. *Atmos Chem Phys* **24**, 2491-2510 (2024).
41. Larsen EH, Moseholm L, Nielsen MM. Atmospheric deposition of trace elements around point sources and human health risk assessment. II: Uptake of arsenic and chromium by vegetables grown near a wood preservation factory. *Science of The Total Environment* **126**, 263-275 (1992).
42. De Temmerman L, Ruttens A, Waegeneers N. Impact of atmospheric deposition of As, Cd and Pb on their concentration in carrot and celeriac. *Environmental Pollution* **166**, 187-195 (2012).
43. Glabonjat RA, Raber G, Van Mooy BAS, Francesconi KA. Arsenobetaine in Seawater: Depth Profiles from Selected Sites in the North Atlantic. *Environmental Science & Technology* **52**, 522-530 (2018).
44. Wurl O, Zimmer L, Cutter GA. Arsenic and phosphorus biogeochemistry in the ocean: Arsenic species as proxies for P-limitation. *Limnology and Oceanography* **58**, 729-740 (2013).
45. Wurl O, Shelley RU, Landing WM, Cutter GA. Biogeochemistry of dissolved arsenic in the temperate to tropical North Atlantic Ocean. *Deep Sea Research Part II: Topical Studies in Oceanography* **116**, 240-250 (2015).
46. Wang P, Sun G, Jia Y, Meharg AA, Zhu Y. A review on completing arsenic biogeochemical cycle: Microbial volatilization of arsines in environment. *Journal of Environmental Sciences* **26**, 371-381 (2014).

Author response to referees' comments (revision 2)

We thank the two reviewers for their time reviewing our manuscript and their valuable comments. Note that the reviewers' comments are written in blue and the authors' responses are written in black. All page numbers given in the answers below refer to the revised manuscript in which changes are indicated using the track changes feature of word processor (i.e., revised manuscript with mark-ups).

Response to Reviewer #1:

The reviewer thanks the authors for a rebuttal on a high level. The reviewer agrees with all answers and would recommend to publish this manuscript in this journal.

We would like to thank the reviewer for the positive feedback on our revised manuscript and again for their previous comments that helped us to improve our manuscript.

Response to Reviewer #2:

The manuscript has been tidied-up by the authors in response to the reviewers wishes to improve its readability and interpretation. The rebuttal was thorough and many points were changed on the manuscript. However, I still have concerns that the dataset is relatively light in numbers of the key precipitation samples, in combination with air masses of clusters already being mixed in land and air contribution, and overlapping.

We agree that our dataset has a limited number of precipitation samples (n=26 samples) and therefore we both analysed precipitation as well as aerosol samples (n=70) for our investigation of potential source contributions. We also agree that contributing moisture sources of precipitation sub-events always have a degree of overlap at our study site (in fact at most study sites). This can for example be seen by the modelled percentages of moisture source contributions presented in Supplementary Table 3. In the revised manuscript, we acknowledged and explained the limitations of the different datasets (precipitation and aerosols) by adding the following sentences:

- *"While we did not use the aerosol samples for back trajectory analyses due the weekly time resolution that would lead to overlapping source contributions, we did analyse them to determine main groups of organic compounds as a further source proxy. It should be noted that these analyses were done on the whole sample (i.e., total pyrolysis of aerosol filter) and not on liquid extracts as were used for As speciation analyses (pages 9, L285-297)*
- *"However, as the number of (sub)events in cluster 4 is low, the mixed source and coastal contributions warrant further investigations." (page 10, L320-322)*
- *"However, it should be noted that Py-GC/MS analysis cannot distinguish between terrestrial, coastal or marine biogenic sources, which could be addressed in follow-up studies." (page 10, L336-338)*
- *"Further field campaigns are needed to confirm these species-specific source contributions with larger numbers of precipitation (sub)events and/or aerosol samples*

collected at high temporal resolution. Furthermore, the extent to which the source contributions of methylated As species vary with sampling locations also warrants further investigation.” (page 14, L488-492)

Furthermore, our conclusions related to potential sources of As species are not exclusively based on moisture source diagnostics for precipitation samples, but also on chemical proxies obtained for both precipitation samples (S species and S isotopes, elemental and DOC content) and aerosol samples (S species, and organic aerosol composition). To clarify why and how moisture source/chemical proxy data (for precipitation sample) as well as precipitation/aerosol datasets were combined, and to improve clarity of the interpretation, we revised the manuscript as follows:

- 1) We revised the section “*Identification of contributing moisture sources and source indicators*” (pages 7-9, L211-297) and associated Figure 3 to provide:
 - A clear description of the moisture sources of precipitation sub-events, as well as of the purpose and output of the hierarchical cluster analysis performed on this dataset (related to Figure 3a,b; page 32)
 - A complete presentation of the variability of all investigated chemical sources proxies in identified moisture sources clusters, with information on significant differences (Mann-Whitney test) (related to Figure 3c-f, page 32)
 - A description of source proxies used for aerosol time series, including detailed description of organic aerosol composition (Supplementary Figure 8), and of the purpose of this aerosol dataset.

- 2) We provided a more complete presentation of the data and of the relationship between source indicators and As species in both precipitation and aerosol samples (see also answer to point 6). For this purpose, we updated figures in the main manuscript (Figures 1, 3, 4 and 7; see pages 28-29, 31-32, 33-34, and 37, respectively) as well as included new figures in the Supplementary Information (Supplementary Figures 4, 8, 9, 10, 11, 14, 15, 16) to visualize all data of precipitation sub-events and the aerosol time series, including scatter plots of all described correlations with As species in the manuscript and corresponding scatter plots for other As species (see answer to detailed points 3 and 6). We revised the text in sections “*TMA^sV^O has mixed terrestrial and marine biogenic sources with likely important coastal contributions*” (pages 9-10, L300-338) and “*DMA^sV^O has a dominant marine origin*” (pages 11-12, L362-399) accordingly. We also re-organized the discussion on sources of TMA^sV^O and DMA^sV^O to improve clarity by moving some information in a new section entitled “*No indications for an atmospheric abiotic source of methylated As species*” (page 12, L409-423).

Here are my points:

1. Cluster 4, on which the interpretation of this paper focused, is labelled “Atlantic/Spain” (Figure 4), yet the authors interpret this as terrestrial. It seems, visually, that the Spain/France cluster 1 has a far greater terrestrial component to it than cluster 4, yet as a lower median TMAO precipitation content?

Our paper does not focus on Cluster 4 given that we present the variability of methylated arsenic species over time and across different atmospheric samples, including precipitation (n=26), cloud water (n=56), and aerosols (n=70; two types of applied extractions). Furthermore, as noted above, our conclusions regarding the potential sources of As species

are not based solely on moisture source diagnostics for precipitation samples (and their associated clusters) but also on relationships with chemical proxies obtained from both precipitation (including S species, S isotopes, elemental composition, and DOC content) and aerosol samples (S species and organic aerosol composition).

We agree that cluster 1 has a stronger terrestrial influence than cluster 4. Cluster 1 (“Spain/France”) has 72% land moisture sources and significantly more land sources than cluster 4 (“Atlantic/ Spain), which has 55% oceanic moisture and 45% land sources (as can be seen in Supplementary Table 3).

Overall, compared to the more terrestrial clusters 1 and 3, cluster 4 has more marine influences (more MSA and higher Na:Sr ratio, Figure 3c,d; page 32) but compared to the Atlantic cluster (cluster 2), cluster 4 has more terrestrial influences, i.e., positive correlation between TMA^{VO} and DOC ($r_s=0.439$, $n=26$, $p<0.05$, Figure 4b; page 34) and a more terrestrial signature in $\delta^{34}\text{S}$ (Figure 3e; page 32). These findings indicate that both the moisture source information and chemical proxies are indicative of mixed marine and terrestrial sources for cluster 4.

Thanks to the comments of the 2nd reviewer, we had a closer inspection of these mixed moisture sources of this cluster. Compared to all other clusters, cluster 4 is not only the most mixed (in terms of oceanic and land sources, based on moisture source diagnostics and chemical proxies, as mentioned above) but also has the largest coastal fingerprint, covering large stretches of coastlines of the Iberian Peninsula. Therefore, for this cluster, we added another broad category in addition to land and marine moisture sources, i.e., a mixed moisture source with likely an important contribution from coastal regions.

It is clear that the few precipitation events that had such mixed marine-land sources with coastal contribution (cluster 4 events) had the highest share of TMA^{VO} proportions, which surpassed those observed in the predominantly land clusters (clusters 1 and 3) and the marine cluster (cluster 2). This observation suggests a potentially important coastal source of TMA^{VO} .

We thus refined our discussion and statements regarding TMA^{VO} sources as follows:

- We wrote in the TMA^{VO} section, titled now “ *TMA^{VO} has mixed terrestrial and marine biogenic sources with likely important coastal contributions*”:

“Significantly higher proportions of TMA^{VO} were found in cluster 4 (Atlantic/Spain), and to a lesser extent in cluster 1 (Spain/France), in comparison to clusters 2 and 3 (Mann-Whitney-U test, $p<0.05$; Fig. 4a). Sources originating from Portugal and Spain seem particularly important for TMA^{VO} according to positive correlations between moisture sources from these regions and TMA^{VO} proportions ($n=26$, $p<0.05$), which was not observed for the other As species (Supplementary Figure 9). The mixed marine and terrestrial moisture source contribution and chemical proxy data suggest that both marine and terrestrial contributions are important for TMA^{VO} occurrences. Compared to the more terrestrial clusters 1 and 3, cluster 4 has more marine influences (more MSA and higher Na:Sr ratio) but compared to the Atlantic cluster (cluster 2), cluster 4 has more terrestrial influences, i.e., positive correlation between TMA^{VO} and DOC ($r_s=0.439$, $n=26$, $p<0.05$, Figure 4b) and a more terrestrial signature in $\delta^{34}\text{S}$. Potential terrestrial sources of TMA^{VO} in atmospheric deposition include volatile emissions of trimethylarsine from soils and peatlands¹², volcanoes¹⁶, as well as sewage sludge⁵⁴ and biogas plants^{16, 55} and potential marine sources are emissions by the marine biosphere^{13, 21, 56}. Based on the mixed signal and important contribution of coastal moisture sources, including the northern, western and south-western coast of the Iberian Peninsula, we suggest that coastal emissions of TMA^{VO} and/or its precursor trimethylarsine could be particularly relevant (see Figure 4c). However, as the number of (sub)events in cluster 4 is low, the mixed source and coastal contributions warrant further investigations.” (page 9, L302-322)

- We revised Figure 4c to indicate this further differentiation into more marine, more terrestrial and mixed source areas with a potentially coastal origin (see pages 33-34).
- We wrote in the last, concluding paragraph of the manuscript: *“While we found indications for a mixed marine and terrestrial biogenic source for TMA_svO, and potentially more coastal sources, for DMA_s^v, we found more distinct marine source indicators (Fig. 7).”* (page 14, L483-486)
- We revised the final concluding Figure 7 to show potential sources of TMA_s^vO from both terrestrial and marine environments (see page 37).

2. I note again that cluster 1 and cluster 4 have only an N of 4, from one sampling campaign, and I would be more convinced by their arguments based on low N if there were further campaigns that showed the same information.

Clusters 1 and 4 indeed have a small number of points. However, for all the correlation plots and further data analyses, the clusters are not individually treated but always combined (as indicated by sample size for each reported correlation coefficients), reflecting the kind of continuum that the clusters together show (also now more clearly indicated by colouring all data points, see answer to the comment below).

To clarify this limitation and acknowledge the need for further studies exploring TMA_s^vO sources in atmospheric deposition, we added the following sentences:

- *“However, as the number of (sub)events in cluster 4 is low, the mixed source and coastal contributions warrant further investigations.”* (page 9, L320-322)
- *“Further field campaigns are needed to confirm these species-specific source contributions with larger numbers of precipitation (sub)events and/or aerosol samples collected at high temporal resolution. Furthermore, the extent to which the source contributions of methylated As species vary with sampling locations also warrants further investigation.”* (page 14, L4587-491)

3. Figure 5c points should have symbols to indicate which clusters they originate from – this would aid interpretation considerably, as the reader could then see if the cluster 4 points were the sole high ones, as from my visual interpretation at least one of these high points is from the Atlantic cluster 2, which somewhat confounds their thesis that DOC is of terrestrial origin, a point one which their thesis is strongly based – see line 258? It would also be highly useful for the reader if 95% CIs for the regression line were included for this graph as the correlation is relatively weak and influenced by a group of 4 points that fall outside another group of data with lower DOC's that has a lot of variance.

We have now clearly indicated the clusters in the figure by colouring them and added a confidence interval for the regression line (Figure 4b; page 34). As indicated in our answer to comment 1 above, we added a separate moisture source class to represent mixed marine and terrestrial sources, likely with high coastal contributions (cluster 4).

For a more complete data presentation, we made the same figures as for TMA_s^vO (i.e., same figure as Figure 4, page 34) also for the other As species (apart from MMA_s^v, which was only detected in a few precipitation sub-events). This enables readers to clearly visualize all

correlations between As species and chemical species as well as between As species and modelled residence time in the continental and Atlantic boundary layers (see Supplementary Figures 14-15).

4. The correlations of arsenic species with specific groups of organic clusters for both TMAO origin seem important, line 268 onwards, yet we are only shown a summary of correlation P values, SI Table 4, with no presentation of the actual data - unless I am mistaken?

We have added the following figures to provide a complete overview of data collected on the aerosol time series (2015-2020):

- Figure 1 in the manuscript shows the temporal variability of As concentrations in the different aerosol extracts: total digest, water extract and 1% HNO₃ extract (page 29)
- Supplementary Figure 4c shows the correlations between the proportions of As species in the aerosol water extracts
- Supplementary Figure 8 shows the temporal variability of organic compound groups identified by Py-GC/MS in the analysed aerosol time series
- Supplementary Figure 16 shows the relationship between the proportions of TMA^s_VO, DMA^s_V and inorganic As (iAs) with methane sulfonic acid (MSA) in the aerosol water extracts.

The complete datasets are also available as in the data repository and as an additional supplementary data file as explained the section "Data Availability".

In addition, we clarified the use of the organic aerosol compositions as source proxy in the section "*Identification of contributing moisture sources and source indicators*" as follows:

"While we did not use the aerosol samples for back trajectory analyses due the weekly time resolution that would lead to overlapping source contributions, we did analyse them to determine main groups of organic compounds as a further source proxy. It should be noted that these analyses were done on the whole sample (i.e., total pyrolysis of aerosol filter) and not on liquid extracts as were used for As speciation analyses. Organic compound groups were identified by Py-GC/MS, a high throughput method that only requires little carbon mass^{51, 52}. As in the few previous studies that used this technique for atmospheric samples, we identified n-alkanes, n-alkenes, (poly)aromatics, and phenols that have been previously linked to anthropogenic sources and urban environments^{48, 52, 53}. These organic compound groups accounted for 6-30, 10-24, 2-12 and 2-7% of the sum peak area of identified groups in analyzed summer and autumn aerosols, respectively (Supplementary Figure 8). We also found large proportions of carbohydrates (3-26%) nitrogen (N) compounds (7-46%), and carboxylic acids (4-42%, Supplementary Figure 8), which are likely of biogenic origin^{48, 52, 53}." (pages 8-9, L285-297)

5. Perhaps the limited data presentation and interpretation is due to the number of parameters collected. It seems to me that their data is ripe for Principle Component Analysis interpretation, with Factor Analysis, and this would help clarify the interrelationship between the clusters, arsenic species and origin proxies such as DOC, organic speciation, elements and isotopes?

We followed the advice of the reviewer and performed a principal component analysis (PCA) on the precipitation dataset including moisture sources, As species, DOC, S species and S isotope composition.

Altogether, the PCA output confirms the relationships between As species and specific moisture source regions and/or chemical source proxies as presented in the main manuscript,

i.e., specifically, the stronger relationship of DMAs^V with indicators for marine sources and, in comparison to DMAs^V, a more terrestrial source signature for TMAs^VO, again indicating its mixed and potential coastal origin.

The PCA output and discussion are presented in Supplementary Figure 11 and Supplementary Note 1, to which we refer to in the main manuscript as follows:

“A principal component analysis performed on the precipitation dataset including moisture source, As species, DOC, S species and S isotope composition, confirms the stronger relationship of DMAs^V with indicators for marine sources, in comparison to TMAs^VO, which has a closer relationship with proxies of terrestrial sources, again indicating its mixed and potential coastal origin, see Supplementary Fig. 11a,b (Detailed description of principal component analysis and results in Supplementary Note 1).” (page 12, L390-399)

6. This brings me back my summary point 8 that while I think the manuscript is based on excellent analysis of hard to get samples, with a well written manuscript, and has the potential of show new insights, the hypothesis drawn from the data are based on both limited cluster N size, and limited data presentation (lack of symbol coding on scatter diagrams, reporting of P values with ought showing correlations). In their rebuttal the authors disagree with me on this point and state that “.. we stand confidently behind ... data analysis and its findings”. I disagree with them and think that more can be done to elucidate their data as outlined above.

In the revised manuscript, we now made three main changes in relation to a further elucidation of our data and a more complete data presentation.

Firstly, we enhanced the data visualization to provide a more comprehensive overview. More “raw” data are now presented in the manuscript (see rFigures 1 and 3, page 28-29 and 31-32, respectively), and scatter plots of all reported correlations in the manuscript are shown in the Supplement Information (e.g., Supplementary Figures 4, 16). Furthermore, for each plot showing any correlations between the various parameters and a specific As species, we provide corresponding plots for the other As species (see Supplementary Figures 9, 10, 14, 15, 16). This approach enables direct comparison of the relationships between each As species and the moisture sources and chemical proxies.

Secondly, we had a closer look at cluster 4 that is characterized by mixed marine and terrestrial moisture sources. We now introduced a separate mixed moisture source class (that has a more coastal footprint) and rewrote our discussion and statement about the source contributions for TMAs^VO. Based on the moisture source information and all the further parameters analysed, and in comparison with DMAs^V, we see clearly a lower contribution of TMAs^VO for the predominantly Atlantic cluster 2 and the highest levels were found in the mixed and more coastally defined cluster 4, whereas the more predominantly terrestrial clusters are also in the lower-medium range. Therefore, based on the complete dataset, we conclude that TMAs^VO has both terrestrial sources (terrestrial fingerprint is clearly seen in the chemical proxies) as well as marine sources, of which a highest contribution may come from coastal regions. For these regions (in Portugal and Spain), it cannot be clearly separated if these are specific land or marine sources, so this warrants further investigation. On the other hand, DMAs^V has a more dominant marine source contribution than TMAs^VO and seemingly less terrestrial and coastal sources. We revised the manuscript in the relevant sections to reflect these findings based on the further data elucidation, as explained in details on our answer to point 1.

Thirdly, as answered above to comment 5, we performed a PCA analysis, which confirms the interpretation as presented in sections “*TMAs^VO has mixed terrestrial and marine biogenic*

sources with likely important coastal contributions” (pages 9-11, L300-338) and “DMAs^V has a dominant marine origin” (pages 11-12, L362-399).

References

1. Mestrot A, Feldmann J, Krupp EM, Hossain MS, Roman-Ross G, Meharg AA. Field Fluxes and Speciation of Arsines Emanating from Soils. *Environmental Science & Technology* **45**, 1798-1804 (2011).
2. Arndt J, Planer-Friedrich B. Moss bag monitoring as screening technique to estimate the relevance of methylated arsine emission. *Science of The Total Environment* **610-611**, 1590-1594 (2018).
3. Michalke K, Wickenheiser EB, Mehring M, Hirner AV, Hensel R. Production of volatile derivatives of metal(loid)s by microflora involved in anaerobic digestion of sewage sludge. *Appl Environ Microbiol* **66**, 2791-2796 (2000).
4. Mestrot A, Xie W-Y, Xue X, Zhu Y-G. Arsenic volatilization in model anaerobic biogas digesters. *Applied Geochemistry* **33**, 294-297 (2013).
5. Savage L, Carey M, Williams PN, Meharg AA. Biovolatilization of Arsenic as Arsines from Seawater. *Environmental Science & Technology* **52**, 3968-3974 (2018).
6. Savage L, Carey M, Williams PN, Meharg AA. Maritime Deposition of Organic and Inorganic Arsenic. *Environmental Science & Technology* **53**, 7288-7295 (2019).
7. Glabonjat RA, Raber G, Van Mooy BAS, Francesconi KA. Arsenobetaine in Seawater: Depth Profiles from Selected Sites in the North Atlantic. *Environmental Science & Technology* **52**, 522-530 (2018).
8. Tolu J, Gerber L, Boily J-F, Bindler R. High-throughput characterization of sediment organic matter by pyrolysis–gas chromatography/mass spectrometry and multivariate curve resolution: A promising analytical tool in (paleo)limnology. *Analytica Chimica Acta* **880**, 93-102 (2015).
9. Chow JC, *et al.* The application of thermal methods for determining chemical composition of carbonaceous aerosols: A review. *Journal of Environmental Science and Health, Part A* **42**, 1521-1541 (2007).
10. Zhao J, Peng Pa, Song J, Ma S, Sheng G, Fu J. Characterization of organic matter in total suspended particles by thermodesorption and pyrolysis-gas chromatography-mass spectrometry. *Journal of Environmental Sciences* **21**, 1658-1666 (2009).
11. Breuninger ES, *et al.* Influences of sources and weather dynamics on atmospheric deposition of Se species and other trace elements. *Atmos Chem Phys* **24**, 2491-2510 (2024).